# Decoupling of dissolved organic matter patterns between stream and riparian groundwater in a headwater forested catchment

Susana Bernal[1,2], Anna Lupon[2,3], Núria Catalán[4], Sara Castelar[1], Eugènia Martí[1]

[1]Center for Advanced Studies of Blanes (CEAB-CSIC), Blanes, 17300, Spain
5  [2]Departament de Biologia Evolutiva, Ecologia i Ciències Ambientals (BEECA), Universitat de Barcelona, Barcelona, 08028, Spain
[3]Departament of Forest Ecology and Management, Swedish University of Agricultural Sciences, Umeå, 90183, Sweden
[4]ICRA, Catalan Institute for Water Research, Girona, 17003, Spain

*Correspondence to*: Susana Bernal (sbernal@ceab.csic.es)

10  **Abstract.** Streams are important sources of carbon to the atmosphere, though whether they merely outgas terrestrially derived carbon dioxide or mineralize terrestrial inputs of dissolved organic matter (DOM) is still a big challenge in ecology. The objective of this study was to investigate the influence of riparian groundwater (GW) and in-stream processes on the temporal pattern of stream DOM concentrations and quality in a forested headwater stream, and whether this influence differed between the leaf litter fall period (LLF) and the remaining part of the year (non-LLF). The spectroscopic indexes (fluorescence index, 15  biological index, humification index, and PARAFAC components) indicated that DOM had an eminently protein-like character and was most likely originated from microbial sources and recent biological activity in both stream water and riparian GW. However, paired samples of stream water and riparian GW showed that dissolved organic carbon (DOC) and nitrogen (DON) concentrations as well as the spectroscopic character of DOM differed between the two compartments throughout the year. A simple mass balance approach indicated that in-stream processes along the reach contributed to reduce DOC and DON fluxes 20  by 50% and 30%, respectively. Further, in-stream DOC and DON uptake were unrelated to each other, suggesting that these two compounds underwent different biogeochemical pathways. During the LLF period, stream DOC and DOC:DON ratios were higher than during the non-LLF period, and spectroscopic indexes suggested a major influence of terrestrial vegetation on stream DOM. Our study highlights that stream DOM is not merely a reflection of riparian GW entering the stream and that headwater streams have the capacity to internally produce, transform, and consume DOM.

## 1 Introduction

The transport of dissolved organic matter (DOM) through fluvial networks is of major importance for understanding the links between continental and coastal biogeochemical cycles (Seitzinger and Sanders, 1997; Battin et al., 2008). Stream DOM is a combination of allochthonous (i.e. terrestrially derived) and autochthonous (i.e. in-stream produced) DOM. The former originates mostly from terrestrial systems (i.e. soils, vegetation and microbes) and it is transported to streams via surface and 30  groundwater flow paths, while the latter derives from in-stream metabolic activity and leachates of litter falling into the stream

especially during the leaf litter period (Qualls and Haines, 1991, 1992). The bioavailability of DOM can differ substantially between allochthonous and autochthonous sources, and thus, a good assessment of the origin and quality of stream DOM is of great importance to understand the capacity of aquatic ecosystems to store and transform carbon (C) and nitrogen (N) (Cole et al., 2007; Battin et al., 2008; Tranvik et al., 2009). Yet, our knowledge of the contribution of allochthonous vs autochthonous sources to total stream DOM and its variability over time and space is far from complete.

The strong correlation found between dissolved organic carbon (DOC) and nitrogen (DON) in temperate and boreal streams have suggested that the soil organic pool is a major factor controlling the fate and form of stream DOM (Perakis and Hedin, 2002; Hedin et al., 1995; Brookshire et al., 2007; Sponseller et al., 2014). These previous observations are the cornerstone of the passive carbon vehicle hypothesis, which states that soil DOM is stoichiometrically static and behaves almost conservatively when travelling throughout the catchment and stream ecosystems (Brookshire et al., 2007). However, there is an increasing body of studies reporting differences in DOC:DON ratios between allochthonous sources and stream water. For instance, stream DOC:DON ratios can change as a consequence of in-stream heterotrophic DOM production during periods of high ecosystem respiration (Caraco and Cole, 2003; Kaushal and Lewis, 2005; Johnson et al., 2013). Moreover, stream biota can show a strong capacity to process DOM (McDowell, 1985; Bernhardt and McDowell, 2008), with whole-reach DOM uptake rates being even higher than for essential nutrients such as nitrate (Brookshire et al., 2005). The processing of DOM within the stream can lead to a decoupling between stream DOC and DON concentrations because stream DOC is mostly used as an energy source, while DON can alternatively be used as a nutrient source (Kaushal and Lewis, 2005; Lutz et al., 2011; Wymore et al., 2015). Therefore, a significant fraction of stream DOM could be degraded, mineralized, or produced within the stream (either in the stream column or in the hyporheic zone).

Despite the potential role of in-stream biota on processing DOM, its ability to modify DOM concentrations and regulate allochthonous DOM fluxes remains elusive. First, the high variety of molecules used during in situ DOM additions (from monomeric carbohydrates to complex leachate molecules) limits the possibility to compare whole-reach DOM uptake rates among sites and to link manipulative experiments with actual DOM processing under natural conditions (Newbold et al., 2006; Bernhardt and McDowell, 2008). Second, the intrinsic complexity of up-scaling reach scale measurements constrains our understanding of the potential of in-stream processes to modify DOM export at catchment scale (Wollheim et al., 2015). Recent synoptic studies suggest that changes in stream DOC concentrations can be mostly explained by hydrological mixing of different water sources, thus suggesting minimal removal of DOC within streams (Tiwari et al., 2014; Wollheim et al., 2015). Yet, these studies are mostly performed during particular periods (usually summer) and in catchments with large wetland and peatland areas that provide large quantities of allochthonous DOM to aquatic ecosystems (Wollheim et al., 2015). Studies with a network perspective are still scarce and usually deal with a high amount of uncertainty because the quantity and quality of DOM in groundwater traversing the hyporheic zone and entering the stream is poorly characterized (Tiwari et al., 2014; Casas-Ruíz et al., 2017).

The objective of this study was to investigate the influence of DOM inputs from riparian groundwater (GW) and in-stream processes on the temporal pattern of stream DOC and DON concentrations and quality (DOC:DON stoichiometry and DOM spectroscopic descriptors) in a Mediterranean forested headwater stream. To do so, we assessed the temporal variation of DOM quantity and quality in stream water and riparian GW along 1.5 years. We expected that differences between riparian GW and

stream DOM would be small if (i) allochthonous sources dominate the temporal pattern of DOM inputs and, (ii) DOM is transported passively along the stream as stated by the carbon vehicle hypothesis (Brookshire et al., 2007). Alternatively, differences between riparian GW and stream water would indicate DOM generation and/or processing of allochthonous DOM within the stream. Specifically, we expected large differences between riparian GW and stream DOM associated with the leaf litter fall period because leachates from fresh material stored in the streambed may increase DOM concentration and fuel

heterotrophic stream metabolism.

## 2 Study Site

The study was conducted from October 2010 to December 2011 in the Font del Regàs catchment (14.2 $km^2$), located in the Montseny Natural Park, NE Spain (41º 50' N, 2º 30' E, 300-1200 m a.s.l.). The climate is sub-humid Mediterranean, with mild winters and dry summers. Mean annual precipitation (975 mm) and temperature (12.9 ℃) during the study period fall within

the long-term annual average for this region (Catalan Metereologic Service: http://www.meteo.cat/servmet/index.html).

The catchment is dominated by biotitic granite and it has steep slopes (28%). Evergreen oak (*Quercus ilex*) and beech (*Fagus sylvatica*) forests cover 54% and 38% of the catchment area, respectively (Fig. 1). The upper part of the catchment (2%) is covered by heathlands and grasslands. Population density within the catchment is <1 person $km^{-2}$. Hillslope soils (pH ~ 6) are sandy and have a 3 cm deep organic layer (O-horizon) followed by a 5 to 15 cm deep mineral layer (A-horizon). The riparian

zone is relatively flat (slope < 10 %), and it covers 6 % of the catchment area. Riparian soils (pH ~ 7) are sandy-loam and they have a 5 cm deep O-horizon followed by a 30 cm deep A-horizon. The width of the riparian zone increases from 6 to 32 m from the upper to the lower part of the catchment, whereas the total basal area of riparian trees increases by 12 fold (Bernal et al., 2015). *Alnus glutinosa*, *Robinia pseudoacacia*, *Platanus hybrida*, and *Fraxinus excelsior* are the most abundant riparian tree species followed by *Corylus avellana*, *Populus tremula*, *Populus nigra*, and *Sambucus nigra*. During base flow conditions,

the riparian GW table is well below the soil surface (~ 50 cm), though it can reach the superficial soil organic layers during storm events (Lupon et al., 2016a).

The catchment is drained by a perennial 3rd order stream. At the headwaters, the streambed is mainly composed of rocks and cobbles (70 %) with a small contribution of sand (~10 %). At the valley bottom, sands and gravels represent 44 % of the stream substrate and the presence of rocks is minor (14 %). During base flow conditions, mean stream water velocity is 0.3 m $s^{-1}$. On

average, stream discharge increases along the reach from 20 to 70 L $s^{-1}$. During the study period, the stream gained water in net terms along the reach, yet it lost water towards the riparian zone in some segments, specifically during summer months.

Moreover, mean area-specific stream discharge decreased longitudinally, an indication that hydrological retention was higher at the valley bottom compared to upstream segments. Permanent tributaries comprise about 50% of the catchment area and contribute 56% of stream discharge (Bernal et al., 2015).

**3 Material and Methods**

**3.1 Field sampling**

We selected 15 sampling sites along a 3.7 km reach that were located from 110 to 600 m apart from each other (Fig. 1). At each sampling site, we installed a 1 m long PVC piezometer (3cm Ø) in the riparian zone (~1.5 m from the stream channel edge). We assumed this water to be representative of the groundwater entering the stream. We collected stream water (from

the thalweg) and riparian GW from each sampling site every 2 months from October 2010 to December 2011. Groundwater samples were collected with a 100 ml syringe connected to a silicone tube. Water samples were collected with pre-acid washed polyethylene bottles after triple-rinsing them with either stream water or groundwater. Field sampling was conducted during base flow conditions to capture the influence in-stream processes on DOM dynamics when they are expected to be the highest. Moreover, by avoiding storm flows, we ensured that riparian GW was the main subsurface water source contributing to stream

runoff. All field campaigns were performed at least nine days after storm events, except for October 2011. At each sampling site, we measured stream discharge ($Q$, in L s$^{-1}$) by adding 1 L of NaCl-enriched solution to the stream (Gordon et al., 2004). The empirical uncertainty associated with $Q$ was calculated considering pairs of measurements conducted under equal water depth conditions as described in Bernal et al. (2015). On each sampling date, we also collected stream water and measured $Q$ at the four permanent tributaries discharging to Font del Regàs stream, which drained 1.9, 3.2, 1.8, and 1.1 km$^2$, respectively

(Fig. 1). These data were used for mass balance calculations (see below).

**3.2 Laboratory analysis and DOM quality indexes**

Water samples were filtered through pre-ashed GF/F filters (Whatman®) and kept cold (< 4 ℃) until laboratory analysis (< 24h after collection). Chloride (Cl$^-$) was used as a conservative hydrological tracer and analyzed by ionic chromatography (Compact IC-761, Metrhom). DOC and total dissolved nitrogen (TDN) concentrations were determined using a Shimadzu

TOC-VCS coupled to a TN analyzer. DOC was determined by oxidative combustion infra-red analysis and TDN by oxidative combustion-chemiluminescence. DON concentration was calculated by subtracting nitrate ($NO_3^-$) and ammonium ($NH_4^+$) concentrations from TDN. Concentrations of $NO_3^-$ and $NH_4^+$ were determined by standard colorimetric methods (details in Bernal et al., 2015).

We used different metrics to assess the quality of DOM and to infer its origin. First, the DOC:DON ratio was used as a general

proxy of DOM quality, high values being indicative of plant organic matter sources (Bernal et al., 2005). Then, we assessed

DOM properties by optical spectroscopy. Fluorescence excitation-emission spectra were recorded on a Shimadzu RF-5301 PC spectrofluorimeter over an emission range of 270-700 nm (1 nm steps) and an excitation range of 230-430 nm (10 nm steps). Measurements were done at room temperature (20-25 ºC) and corrected for instrument baseline offset. A Milli-Q blank was subtracted from each sample to eliminate Raman scattering.Sampling blanks were included to assess for leaching of DOM

during the sampling procedure. We followed the procedure in Kothawala et al. (2013) for inner filter correction. Briefly, UV-Vis absorbance spectra (200-800 nm) were obtained in a Shimadzu UV-1700 spectrophotometer, using 1 cm quartz cuvette. Due to fatal circumstances, absorbance spectra could not be recorded for some samples. In these cases, we used the modeled mean absorbance spectra for either riparian GW or surface stream water to apply the inner filter correction. All the corrections were applied using the FDOM correct toolbox for MATLAB (Mathworks, Natick, MA, USA) following Murphy et al. (2010).

We calculated three spectroscopic descriptors: (i) the fluorescence index (*FI*) which typically ranges from ~1.2 to ~2 and is linked to the DOM origin with low values being characteristic of terrestrial higher-plant DOM sources and high values of microbial DOM sources (Jaffé et al., 2008), (ii) the biological index (*BIX*), for which higher values indicate a higher contribution of recently produced DOM (i.e. biological activity or aquatic bacterial origin) (Huguet et al., 2009), and (iii) the humification index (*HIX*) as a proxy of the humification status of DOM (i.e. higher values indicating higher humification

degree) (Ohno, 2002; Fellman et al., 2010).

Parallel Factor Analysis (PARAFAC) was used to identify the main fluorescence components of DOM (Stedmon et al., 2003). The analysis was performed using the DrEEM toolbox for MATLAB (Mathworks, Inc., Natick, MA) according to Murphy et al. (2013). Scatter peaks and outliers were removed and samples normalized to its total fluorescence prior to fitting the PARAFAC model. The appropriate number of components was determined by visual inspection of both the residual

fluorescence and the components behavior as organic fluorophores. The PARAFAC modeling of EEM spectra from the analyzed samples revealed four independent components (F1-F4; Fig. S1 in Supplementary Information). Components F2 and F3 corresponded to humic-like materials, while components F1 and F4 to protein-like fluorescence (Table S1 and S2). The four components model was validated by split-half analysis and random initialization with 10 iterations. Finally, the level of coincidence of the obtained model against other PARAFAC models published in the online repository OpenFluor data base

(http://www.openfluor.org; June 2017) was assessed applying a Tucker congruence coefficient of 95 % (Murphy et al., 2014).

### 3.3 Whole-reach net DOM uptake rates

We investigated the influence of in-stream biogeochemical processes on stream DOM fluxes by applying a mass balance approach for the whole reach. Briefly, we calculated the net flux resulting from in-stream gross uptake and release along the reach (*U*, in µg m$^{-2}$ s$^{-1}$) by including all hydrological input and output solute fluxes (upstream-most site, tributaries, and riparian

GW) in the mass balance. Riparian GW must transverse the hyporheic zone before arriving to the stream water column, and thus, we considered that in-stream net uptake was the result of biogeochemical process occurring in both the stream water column and the hyporheic zone. For each sampling date, *U* for either DOC or DON was approximated with:

$$U = (Q_{top} \times C_{top} + \sum_{i=1}^{4} Q_{tr,i} \times C_{tr,i} + \sum_{j=1}^{14} Q_{gw,j} \times C_{gw,j} - Q_{bot} \times C_{bot})/A, \tag{1}$$

where $Q_{top}$ and $Q_{bot}$ are the discharge at the top and at the bottom of the reach, $Q_{tr}$ is the discharge from tributaries, and $Q_{gw}$ is

the net riparian GW inputs (all in L s$^{-1}$). $Q_{gw}$ was estimated as the difference in $Q$ between consecutive sampling sites and could be either positive (net gaining) or negative (net losing) (Covino et al., 2010). Top and bottom fluxes were calculated by multiplying $Q$ by stream water solute concentration at the top ($C_{top}$) and at the bottom ($C_{bot}$) of the segment, respectively. For each stream segment $j$, riparian GW fluxes were estimated by multiplying $Q_{gw}$ by solute concentration ($C_{gw}$) as described in Bernal et al. (2015). Briefly, $C_{gw}$ averaged riparian GW concentration at the top and bottom of the segment for net gaining

segments ($Q_{gw} > 0$), while it averaged stream water concentrations at the top and bottom of the segment for net losing segments ($Q_{gw} < 0$). For each tributary $i$, the input flux to the stream was calculated by multiplying $Q_{tr}$ and solute concentrations ($C_{tr}$) at the outlet of the tributary. The total active streambed ($A$) was 8860 m$^2$ and it was estimated by multiplying the total length of the reach (3.7 km) by the mean wetted width (2.4 m) that varied < 10% across the different sampling dates. The values used to calculate $U$ for each sampling date are detailed in Table S3. Finally, we calculated an upper and lower limit of $U$ based on

the empirical uncertainty associated with discharge measurements ($Q$ and $Q_{gw}$) (Bernal et al., 2015).

The mass balance approach used in the present study was similar to that applied for Cl$^-$, NH$_4^+$, and NO$_3^-$ for the same study reach and period in Bernal et al. (2015). We considered Cl$^-$ as a hydrological reference because this conservative tracer showed $U \sim 0$ for the whole study period (Bernal et al., 2015). For DOC and DON, U > 0 indicates that gross uptake prevails over release, U < 0 indicates that release prevails over gross uptake, and U ~ 0 indicates that gross uptake ~ release. Therefore, we

expected $U \neq 0$ if DOM does not behave conservatively and in-stream gross uptake and release processes do not fully counterbalance each other. We assumed that $U$ was indistinguishable from 0 when the range of upper and lower limits contained zero.

To assess the contribution of in-stream net uptake to stream DOM fluxes, we calculated the ratio between $U \times A$ (absolute value) and the total input flux ($F_{in}$) for each compound (i.e. DOC and DON) and sampling date. $F_{in}$ was the sum of fluxes

from upstream ($Q_{top} \times C_{top}$), tributaries ($Q_{tr} \times C_{tr}$), and riparian GW ($Q_{gw} \times C_{gw}$). The later was included in the calculation only when the main stream was gaining water in net terms (i.e. $Q_{gw} \times C_{gw} > 0$). We interpreted a high $|U \times A|/F_{in}$ ratio as a strong potential of in-stream processes to modify input fluxes (either as a consequence of gross uptake or release). The relative importance of in-stream DOM uptake and release was estimated with $U > 0/F_{in}$ and $|U < 0|/F_{in}$, respectively. In addition, we calculated the contribution of upstream ($Q_{top} \times C_{top}/F_{in}$) and tributary ($Q_{tr} \times C_{tr}/F_{in}$) inputs to stream DOM fluxes.

**3.4 Statistical analysis**

The data set was divided in two groups based on the temporal pattern of leaf litter fall because we expected large differences between riparian GW and stream DOM associated with the input of fresh leaf litter to the stream. During the two water years,

leaf litter fall began in early October and peaked in early November. In 2010, the litter fall period finished in late November, while it lasted until late December in 2011. There were four sampling dates within the leaf litter fall period (hereafter, LLF) and six sampling dates during the remaining part of the year (hereafter, non-LLF). Median values for each sampling date were used for analyzing the seasonal pattern of stream DOM concentration and quality (DOC:DON ratio and spectroscopic descriptors). We used a Mann Whitney test to analyze differences in DOM concentrations and quality between the LLF and non-LLF periods for both stream water and riparian GW (Zar, 2010). Moreover, we used linear regression models to investigate (i) longitudinal patterns of $Cl^-$ and DOM concentrations, and (ii) differences in DOM stoichiometry (i.e. the relationship between DOC and DON concentration) between riparian GW and stream water.

We explored the influence of riparian GW on the temporal pattern of stream DOM by analyzing the difference between DOM concentrations in these two water compartments with a Wilcoxon paired rank sum test. Tests were run separately for the LLF and non-LLF periods. Moreover, we compared the temporal variation of longitudinal trends in DOM spectroscopic descriptors between stream water and riparian GW. Longitudinal trends were analyzed by applying linear regression and the standardized regression coefficient ($r$) was used as a measure of the strength of the longitudinal pattern along the reach. For a particular sampling date, we expected similar longitudinal trends between stream water and riparian GW (and thus similar $r$) if riparian GW was a major source of DOM to the stream and in-stream processes had a small influence of DOM quality.

Finally, we explored differences in $U$ between LLF and non-LLF periods with a Mann Whitney test. Moreover, we used Spearman's $\rho$ correlations to test (i) whether $U_{DOC}$ and $U_{DON}$ followed the same temporal pattern, and (ii) whether they were behaving conservatively, and thus, similar to $U_{Cl}$.

We chose non-parametric tests for comparing groups of data because the residuals of variables were not always normally distributed (Zar, 2010). All statistical tests were run with JMP v.5.0 statistical software (SAS Institute, Cary, NC).

## 4 Results

### 4.1 Temporal pattern of chloride and DOM in stream water

During the study period, median $Cl^-$ concentration in the main stream was higher for the LLF (8.6 [7.8, 13.1] [25th, 75th percentiles] mg $L^{-1}$) than for the non-LLF period (7.8 [7.3, 8.8] mg $L^{-1}$) (Mann Whitney test, $Z = 2.82$, df $= 1$, $p = 0.005$). Stream $Cl^-$ concentrations increased along the reach by 43% and 48% during the LLF and the non-LLF period, respectively (Fig. 2a). A similar pattern was exhibited by riparian GW (Fig. S2). In the tributaries, median stream $Cl^-$ concentration was 10.2 [8.8, 14.2] mg $L^{-1}$. For DOC, median concentration in the. main stream was higher for the LLF (843 [643, 1243] µg C $L^{-1}$) than for the non-LLF period (406 [304, 580] µg C $L^{-1}$) (Mann Whitney test, $Z = 2.55$, df $= 1$, $p = 0.008$) (Fig. 3a). Stream DOC concentrations increased along the reach by 58% during the LLF period (Fig. 2b). In the tributaries, median DOC concentration was 577 [390, 881] µg C $L^{-1}$. For DON, median concentration in the main stream was 58 [35, 78] µg N $L^{-1}$ and showed no seasonal pattern (Mann Whitney test, $Z = -0.85$, df $= 1$, $p > 0.05$) (Fig. 3b). Stream DON concentrations showed no clear

longitudinal changes for any of the two study periods (Fig. 2c), though concentrations could vary by 40% on a single date. No clear longitudinal pattern was found for either DOC or DON in riparian GW (Fig. S2). In the tributaries, median DON concentration was 54 [34, 75] µg N $L^{-1}$. The median DOC:DON ratio in the main stream was higher during the LLF (DOC:DON = 22 [14, 43]) than during the non-LLF period (DOC:DON = 8 [5, 15]) (Mann Whitney test, $Z = 1.98$, df = 1, $p = 0.033$) (Fig. 3c).

Median values of *FI* ($> 2$) were typical of microbial DOM sources, while low values of *HIX* ($< 2$) indicated that the humification of the samples was low (Fig. 3). Regarding the PARAFAC model, the components F1 and F4 (associated with protein-like materials) were responsible for most of the total fluorescence of stream water samples (50 [46, 53] % and 25 [24, 28] %, respectively). The components F2 and F3 (associated with humic-like materials) accounted for 13 [11, 15] % and 11 [9, 13] % of the total fluorescence, respectively (Fig. 4).

There were differences in stream DOM quality between the LLF and non-LLF period, though most of the spectroscopic metrics (*BIX*, *HIX*, F1, F2, and F4) were similar between the two periods (in the five cases, Mann Whitney test, $p > 0.05$). In contrast, values of *FI* and the humic-like component F3 were higher during the LLF than during the non-LLF period (in the two cases, Mann Whitney test, $Z < 2.24$, df = 1, $p < 0.05$). The relative contribution of F3 to the total fluorescence was higher during the LLF than during the non-LLF period (Mann Whitney test, $Z = 3.43$, df = 1, $p < 0.0006$), while the protein-like component F4 showed the opposite pattern (Mann Whitney test, $Z = -2.23$, df = 1, $p < 0.025$).

## 4.2 Temporal pattern of chloride and DOM in riparian GW

During the study period, median $Cl^-$ concentrations in riparian GW was higher for the LLF (9.8 [7.8, 13.7] mg $L^{-1}$) than for the non-LLF period (8.7 [7.4, 10.6] mg $L^{-1}$). DOC in riparian GW showed a similar pattern, with median concentration higher for the LLF (1411 [1133, 2311] µg C $L^{-1}$) than for the non-LLF period (864 [626, 1414] µg C $L^{-1}$) (Mann Whitney test, $Z = 5.49$, df = 1, $p < 0.001$). In contrast, median DON concentrations in riparian GW were lower during the LLF (67 [45, 157] µg N $L^{-1}$) than during the non-LLF (113 [64, 195] µg N $L^{-1}$) (Mann Whitney test, $Z = -1.96$, df = 1, $p = 0.049$). Riparian GW showed higher DOC:DON ratios during the LLF (DOC:DON = 27 [14, 43]) than during the non-LLF period (DOC:DON = 10 [6, 14]) (Mann Whitney test, $Z = 4.98$, df = 1, $p < 0.001$).

Similar to stream samples, the PARAFAC components related to the protein-like fluorescence (F1 and F4) were responsible for the major part of the total fluorescence of riparian GW samples (44 [38, 49] % and 26 [23, 29] %, respectively). The fluorescence components associated with humic-like materials (F2 and F3) accounted for 16 [13, 21] % and 12 [9, 17] %, respectively.

Values of *FI*, *BIX,* and *HIX* in riparian GW showed no differences between the LLF and non-LLF period, with medians equaling to 2.49 [2.41, 2.61], 0.67 [0.61, 0.74], and 1.11 [0.85, 1.68], , respectively (for the three indexes: Mann Whitney test, df = 1, $p > 0.05$). Regarding PARAFAC, three out of the four fluorescence components (F1, F3, and F4) showed higher values in riparian GW during the LLF than during the non-LLF period (for the three components: Mann Whitney test, df = 1, $p <$

0.015). However, the relative contribution of the four components to the total fluorescence did not change between the two periods (for the four components: Mann Whitney test, df = 1, $p > 0.05$).

## 4.3 Influence of riparian GW on stream DOM

The paired test comparing stream water and riparian GW samples collected simultaneously along the study reach showed that Cl⁻ concentrations were similar between riparian GW and stream water during the LLF period, but higher in the former than in the later during the non-LLF period (Table 1). DOC and DON concentrations were higher in riparian GW than in stream water during both the LLF and the non-LLF period (Table 1). However, there were no differences in DOC:DON ratios between riparian GW and stream water in any of the two periods. During the LLF period, concentrations of DOC and DON were uncorrelated to each other, while stream water and riparian GW showed a positive relationship between DOC and DON concentrations during the non-LLF period (Fig. 5).

Spectroscopic descriptors also show differences between the two water bodies, yet those differences were not consistent between the two study periods. During the LLF period, the *FI* was higher in stream water than in riparian GW, while the opposite trend was observed for indexes associated with both humic-like substances (*HIX* and F2) and in-situ produced, protein-like compounds (*BIX* and F4) (Table 1). During the non-LLF period, *HIX*, F2, F3, and F4 were lower in stream water than in riparian GW, while no differences between the two water bodies were observed for *FI*, *BIX*, and F1 (Table 1).

The longitudinal trends in DOM quality differed between stream water and riparian GW. Values of *FI* in stream water increased along the reach in eight out of 10 sampling dates, while values of *HIX* did so in four out of 10 cases (r > 0 in Fig. 6). Longitudinal trends in stream DOM spectroscopic properties were observed during both the LLF and non-LLF period. In contrast, riparian GW showed no significant longitudinal patterns for either *FI*, *BIX,* or *HIX* in any of the sampling dates. Regarding PARAFAC components, both stream water and riparian GW showed significant changes along the reach in some particular sampling dates. The most consistent pattern was the longitudinal increase in humic-like components (F2+F3), which was observed in four out of 10 sampling dates (Fig. S3).

## 4.4 Contribution of catchment water sources and in-stream processes to stream DOM fluxes

Riparian GW was the most important source of DOC along the reach (58% of the total inputs), while upstream sources provided most of the DON to the stream (30% of the total inputs) (Table 2). The contribution of tributaries to stream DOM fluxes was relatively small compared to stream Cl⁻ fluxes (Table 2).

Values of $U > 0$ were measured for both DOC and DON, indicating that in-stream processes influenced stream DOM fluxes at Font del Regàs. During the study period, median values of $U_{DOC}$ were 197.7 [58.3, 315] µg C m⁻² h⁻¹, whereas values of $U_{DON}$ were 22.3 [4.6, 44.3] µg N m⁻² h⁻¹. Differences in the contribution of in-stream processes to stream DOM fluxes between the LLF and the non-LLF period were not statistically significant (for both $U_{DOC}$ and $U_{DON}$, $Z > Z_{0.05}$, df = 1, p > 0.05). At reach scale, $U$ contributed to modify stream fluxes ($|U x A|/F_{in}$) by 32 [19, 46] % for DOC and 40.5 [29, 52] % for DON. These

values were 10 fold higher than for Cl$^-$ (the conservative tracer), for which $U_{Cl}$ represented 3.6 [1.9, 9.4] % of the input fluxes (Fig. 7a). The stream acted as a net sink of DOM ($U > 0$) in six and seven out of 10 sampling dates for DOC and DON, respectively. In these cases, in-stream processes contributed to reduce stream fluxes by 47 [43, 65] % and 37 [28, 40] % for DOC and DON, respectively (Fig. 7b and c, bars).

There was no significant relationship between $U$ for the different compounds considered in this study. No correlation was found between $U_{Cl}$ and either $U_{DOC}$ or $U_{DON}$ (in both cases: $\rho < 0.3$, $p > 0.05$), indicating that both DOC and DON behaved differently than expected from a conservative tracer. Moreover, $U_{DOC}$ and $U_{DON}$ were unrelated to each other (Fig. 8a).

## 5 Discussion

The capacity of streams to mineralize allochthonous DOM, and thus their ability to contribute to the net balance between C
storage and emission at global scales, remains elusive and available results are contradictory. Most of the uncertainties associated with the estimation of biogeochemical processing rates at large scales (reaches > 100 m) rely on the fact that GW inputs are rarely measured (Tiwari et al., 2014; Casas-Ruíz et al., 2017). Our synoptic approach is unique in the sense that explicitly considers GW inputs, allowing for more reliable C and N budget calculations (Bernal et al., 2015). However, the characterization of the exact DOM chemistry entering from the riparian GW to the stream is a complex issue (e.g. Brookshire
et al., 2009). First, the two water bodies (stream and riparian GW) are hydrologically connected throughout the hyporheic zone (Bencala et al., 2011). Thus, hydrological mixing cannot be completely rule out because stream water can eventually penetrate towards the riparian zone (Bernal et al., 2015). Second, DOM in riparian GW is likely processed while traversing the near-stream and hyporheic zones (Fasching et al., 2015). Hence, by sampling only riparian GW (2 m from the stream channel) and free flowing water at the thalweg, we could not distinguish whether in-stream processes occurred in the stream water column,
the streambed, or the hyporheic zone. Another keen aspect of our study is that we characterized the spectroscopic properties of DOM in both stream water and riparian GW to investigate whether stream DOM reflected allochthonous sources or if in-stream processes modified DOM quality.

Our study highlights that DOM in the Font del Regàs stream and riparian GW had an eminently protein-like character, most likely originated from microbial sources and recent biological activity. For instance, the fluorescence of the samples was
dominated by F1 and F4 (up to 75% of the total fluorescence), two PARAFAC components that presented wavelengths typically attributed to tyrosine and tryptophan (Fellman et al., 2010) (Table S1). Moreover, the whole range of *BIX* values measured in water samples (from 0.4 to 1.63) depicted a strong influence of autochthonous DOM sources (Huguet et al., 2009), while all measured *HIX* values were < 6, indicating low humification of the samples (Fellman et al., 2010). These values contrast with those reported for stream water samples from boreal and temperate catchments with large peatlands and wetland
areas, which usually have high DOC concentrations (> 10 mg C L$^{-1}$) and highly colored humic materials (e.g. Kothawala et al., 2016). However, similar values of both *BIX* and *HIX* to the ones presented here have been reported previously in systems

with low DOC concentrations and not very colored DOM, such as ground caves and spring waters (Birdwell and Engel, 2010; Simon et al., 2010) as well as in soils (Traversa et al., 2014) and some rivers (Huang et al., 2015).

## 5.1 Empirical evidence of in-stream DOM processing

We found that stream DOM did not exhibit a conservative behavior because the stream showed a large capacity to change DOM fluxes (by 30-40%) compared to Cl⁻ fluxes (by 3%). The predominant protein-like character of stream DOM at Font del Regàs could explain, at least partially, why $U_{DOC}$ and $U_{DON}$ differed from zero during most of the study period. This result indicates that in-stream DOM uptake and release processes were not counterbalancing each other (otherwise $U$ would approach zero). For both DOC and DON, we found that in-stream uptake usually predominated over release (i.e. $U > 0$), suggesting higher DOM consumption than production. Our mass balance calculations indicated that in-stream processes could decrease reach scale fluxes up to 80 % and 50 % for DOC and DON, respectively. These findings imply that biogeochemical processes occurring within the stream were able to modify DOC and DON concentrations and fluxes to downstream ecosystems, contrasting with results reported in previous studies (Temnerud et al., 2007; Tiwari et al., 2014; Wollheim et al., 2015). Yet, our results are representative of base flow conditions, which represent ca. 60 % of the annual DOC and DON flux in the study catchment (unpublished data). Moreover, mean water residence time along the reach was relatively low (4h, unpublished data) because running waters predominated and there were no natural or artificial dams. Further studies including storm flow conditions and/or reaches with small reservoirs would be needed to gain a more complete picture of the role of in-stream processes on DOM dynamics and whether headwater streams shifts from reactors to pipes with changing hydrological conditions (Casas-Ruíz et al., 2017; Raymond et al., 2016).

Noteworthy, median values of in-stream net uptake ($U_{DOC}$ = 198 µg C m⁻² h⁻¹ and $U_{DON}$ = 22.3 µg N m⁻² h⁻¹) were 10-1000 fold lower than rates of in-stream gross uptake and DOM production reported for DOM addition experiments in other headwater streams (Lush and Hynes, 1978; McDowell, 1985; Maranger et al., 2005; Bernhardt and McDowell, 2008; Johnson et al., 2013). These discrepancies could be partially explained by the fact that some of these manipulative experiments used monomeric carbohydrates that are easily bioavailable (Mineau et al., 2016). Moreover, and as previously reported for nutrients, differences between estimates of in-stream gross and net uptake suggest that DOM consumption and production likely occur simultaneously within the stream, and that the former is counterbalanced to some extend by the latter (von Schiller et al., 2015). Supporting this idea, median values of $U_{DOC}$ were >100 fold lower than DOC consumption inferred from measurements of ecosystem respiration calculated from diel cycles of dissolved oxygen concentrations in the same study stream (Lupon et al., 2016b).

The observed differences in the spectroscopic properties of DOM between the stream and riparian GW further support the existence of an autochthonous source of labile DOM in the Font del Regàs stream. For instance, riparian GW presented higher humic-like fluorescence (i.e. higher values of *HIX*, F2 and F3) than stream water, which is in agreement with a recent study comparing stream and groundwater DOM (Huang et al., 2015). Moreover, the contribution of the protein-like component F1

to the total fluorescence was higher in stream water (50.6 %) than in riparian GW (43.9 %), while the contribution of F2, a ubiquitous humic component related with fulvic acids and re-processed humics, was higher in riparian GW (17.8 %) than in stream water samples (13.1 %). Finally, the lack of longitudinal trends in DOM quality in riparian GW contrasted with the consistent increase in *FI* observed for stream water along the reach (in eight out of 10 sampling dates). This finding suggests that stream DOM shifted towards a more microbial origin as moved downstream, and that this change was more related to in-stream processes than to changes in the spectroscopic character of riparian GW. Altogether, our results highlight that in-stream processes have the potential to change not only the quantity, but also the quality of DOM, which reinforces their potential role as bioreactors rather than C chimneys transforming dissolved inorganic carbon from terrestrial groundwater to $CO_2$ (Hotchkiss et al., 2015).

## 5.2 Decoupling between in-stream DOC and DON dynamics

We found that the contribution of tributaries to stream DOM fluxes was relatively small (from 10 to 30%) compared to stream $Cl^-$ fluxes (>50%), suggesting that other sources of DOM within the catchment were more important than tributaries. However, dominant catchment sources differed between DOC and DON: riparian GW was the major contributor of DOC, while most of the DON inputs came from upstream. These differences could be partially explained by changes in vegetation: the upstream sites had no riparian zone and drained beech forests exhibiting low mineralization and nitrification rates (Lupon et al., 2016c), while most of the mid- and down-stream sites along the reach were flanked by a well-developed riparian forest that hold higher soil N processing rates (Lupon et al., 2016c).

Despite variances in DOM sources, differences in DOC:DON ratios between stream water and riparian GW were small throughout the year. Moreover, water samples showed a positive and moderate relationship between DOC and DON concentrations, especially during the non-LLF period. Similar DOM stoichiometry between terrestrial and aquatic ecosystems has been typically understood as an indication of the recalcitrant and allochthonous nature of organic matter in stream waters (Perakis and Hedin, 2002; Rastetter et al., 2005). Therefore, these results could a priori suggest that allochthonous DOM inputs mostly dominated DOM in stream water. Yet, the spectroscopic analysis clearly indicated that the quality of DOM differed between these two compartments, and that stream DOM was likely highly available to biota given the high content of protein-like material, which was higher than in riparian GW entering the stream.

In concordance with the idea that stream DOM was not recalcitrant, we found (i) that *U* differed from zero for both DOC and DON, and (ii) that $U_{DOC}$ and $U_{DON}$ were unrelated to each other. This finding supports the hypothesis that these two compounds undergo different metabolic and biogeochemical pathways (Kaushal and Lewis, 2005; Lutz et al., 2011): DOC is mostly used as an energy source, while evidence is growing that DON can also be used as a nutrient (Wymore et al., 2015). The dual behavior of DON could partially explain why $U_{DON}$ was unrelated to $U_{DOC}$, which contrasts with the strong relationship exhibited by in-stream net uptake rates for the two inorganic forms of N, $U_{NO3}$ and $U_{NH4}$, which are both essential nutrients for biota (Fig. 8b). For DOC, a major fraction of what is taken up (~ 70%) follows catabolic pathways (respiration) and is

removed to the atmosphere, while the remaining part (~ 30%) may be used for microbial growth (del Giorgio and Cole, 1998). Thus, considering that in-stream DOM uptake contributed to reduce allochthonous DOC fluxes by 36-54 % (25th and 75th percentiles), approximately one quarter (21-32 %) of the DOC entering or produced within the stream could be released as $CO_2$ to the atmosphere.

**5.3 Influence of leaf litter fall on stream DOM dynamics and spectroscopic properties**

Previous studies have reported large increases in stream DOC concentration and ecosystem respiration associated with large inputs of fresh leaf litter in autumn (e.g. Acuña et al., 2004). Thus, we expected large differences in stream DOM concentrations and quality between the LLF and non-LLF period, as well as between riparian GW and stream DOM during the LLF period. Concordantly, the highest stream DOC concentrations and DOC:DON ratios were measured during the LLF period (specially

in November 2010). Yet, the same pattern was observed for riparian GW, where concentrations of DOC during the LLF period were even higher than in the stream. In this case, higher DOC concentrations could be explained by increases in the groundwater table after autumn rains, which then flow through more superficial organic soil layers (Guarch-Ribot and Butturini, 2016). This idea is supported by the fact that riparian GW showed higher fluorescence during the LLF than the non-LLF period, but no changes in the relative contribution of the four fluorescence components to the total fluorescence. In

contrast, the relative contribution of F3 (humic-like component) and F4 (protein-like component) increased and decreased, respectively, in stream water during the LLF period. This result, together with the higher values of *FI*, bears the idea that leaf litter inputs were a source of humic-like material, but that, at the same time, were fueling microbial activity within the stream. The fact that DOM uptake predominated over release ($U_{DOC}$ and $U_{DON} > 0$) even during some sampling dates within the LLF period supports the hypothesis that fresh particulate organic matter was processed en route and that stream biota was consuming

DOM (Battin et al., 2008; Fasching et al., 2014).

**6 Conclusions and future direction**

Global studies highlight that streams and rivers are important sources of C to the atmosphere (Cole et al., 2007; Raymond et al., 2013). Yet, the potential role of streams to mineralize allochthonous DOC and its consequences at the catchment scale is still largely unknown (Hotchkiss et al., 2015). Our study sheds new light into this issue by showing that headwater streams

have a strong capacity to internally produce, transform, and consume DOM. The mass balance calculations revealed that in-stream processing substantially modified stream DOC and DON fluxes during base flow conditions. Moreover, we found that DOM concentration and spectroscopic character differed between stream water and riparian GW, which provides evidence that stream DOM is not merely a reflection of riparian DOM entering the stream. On the contrary, our findings suggest that both riparian leaf litter inputs and in-stream DOM cycling are essential controls of DOM dynamics in forested headwater

streams. Further work is needed for disentangling the different mechanism underlying DOC and DON processing within the

streams as well as for understanding how environmental factors such as nutrient availability and water residence time drive in-stream DOM processing and changes in DOM quality during different hydrological conditions.

**Data availability**

The data sets used in this paper can be obtained from the authors upon request.

**Authors contribution**

Susana Bernal designed the experiment. Susana Bernal and Anna Lupon carried it out. Sara Castelar, Anna Lupon and Núria Catalan performed all laboratory analysis. Susana Bernal, Anna Lupon, and Núria Catalan analyzed the data set. Susana Bernal prepared the manuscript with contributions from Anna Lupon, Núria Catalan, Sara Castelar and Eugènia Martí.

**Competing interests:** the authors declare that they have no conflict of interest.

**Acknowledgements**

We thank Aitana Oltra for assisting with GIS, Francesc Sabater, Sílvia Poblador, Miquel Ribot, Eduardo Martín, and Clara Romero for field assistance, and Montserrat Solé and Alba Guarch for helping with spectrofluorometric analysis. Susana Bernal and Anna Lupon were funded by the Spanish Ministry of Economy and Competitiveness (MINECO) with a Juan de la Cierva contract (JCI-2010-06397) and a FPU grant (AP-2009-3711). Núria Catalán held a Juan de la Cierva (FJCI-2014-
23064) and a Beatriu de Pinós (BP-2016-00215) grants. Susana Bernal received additional funds from the Spanish Research Council (CSIC) (JAEDOC027) and the MINECO-funded projects MED_SOUL (CGL2014-59977-C3-2-R) and NICUS (CGL-2014-55234-JIN). The Vichy Catalan Company, the Regàs family and the Catalan Water Agency (ACA) graciously gave us permission for sampling at the Font del Regàs catchment.

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

**Figures**

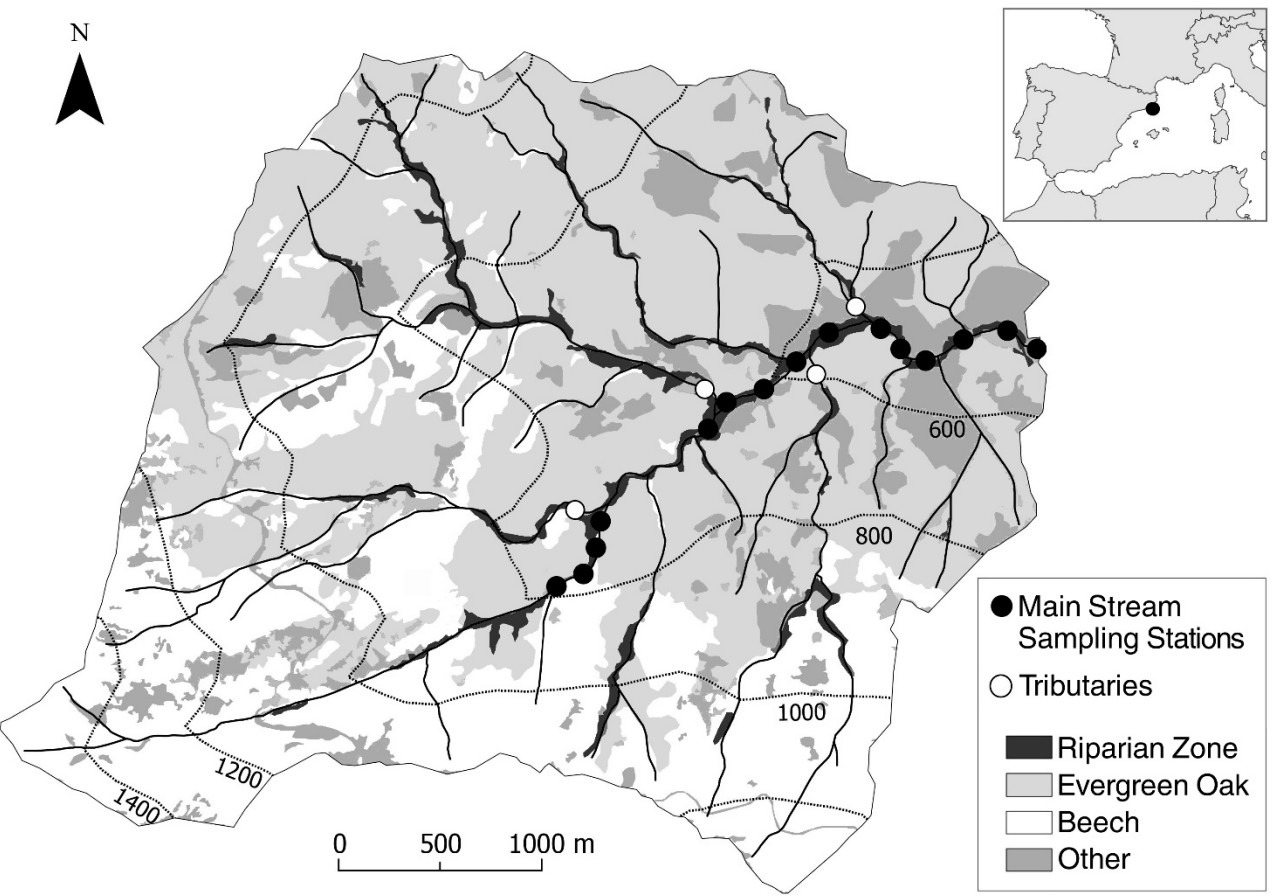

5    **Figure 1. Map of the Font del Regàs catchment within the Montseny Natural Park (NE, Spain). The vegetation cover and the main stream sampling stations along the 3.7km reach are indicated. Four permanent tributaries discharged to the main stream from the upstream- to the downstream-most site (white circles). The remaining tributaries were dry during the study period.**

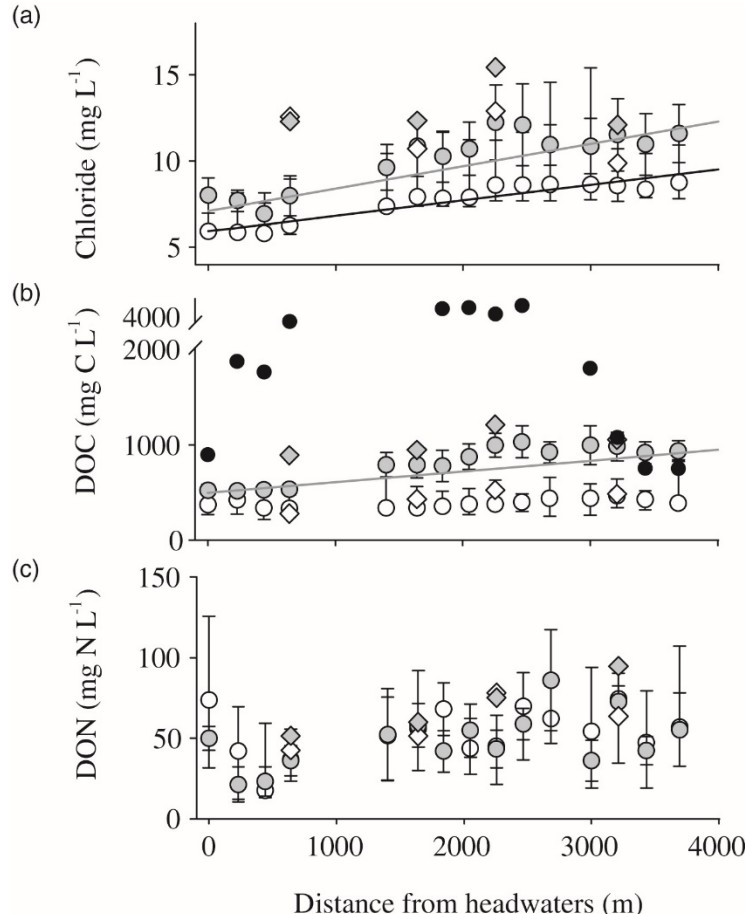

**Figure 2. Longitudinal patterns of (a) chloride, (b) dissolved organic carbon (DOC), and (c) dissolved organic nitrogen (DON) concentrations in stream water along the 3.7km reach. Symbols are median values and whiskers are the interquartile range (25th, 75th percentiles) for the main stream (circles) and tributaries (diamonds). Concentrations are shown separately for the LLF (grey) and non-LLF period (white). Black circles in (b) correspond to the field campaign of November 2010 when DOC concentrations were higher than for the remaining study period. Model regressions are indicated with solid lines only when significant (tributaries not included in the model).**

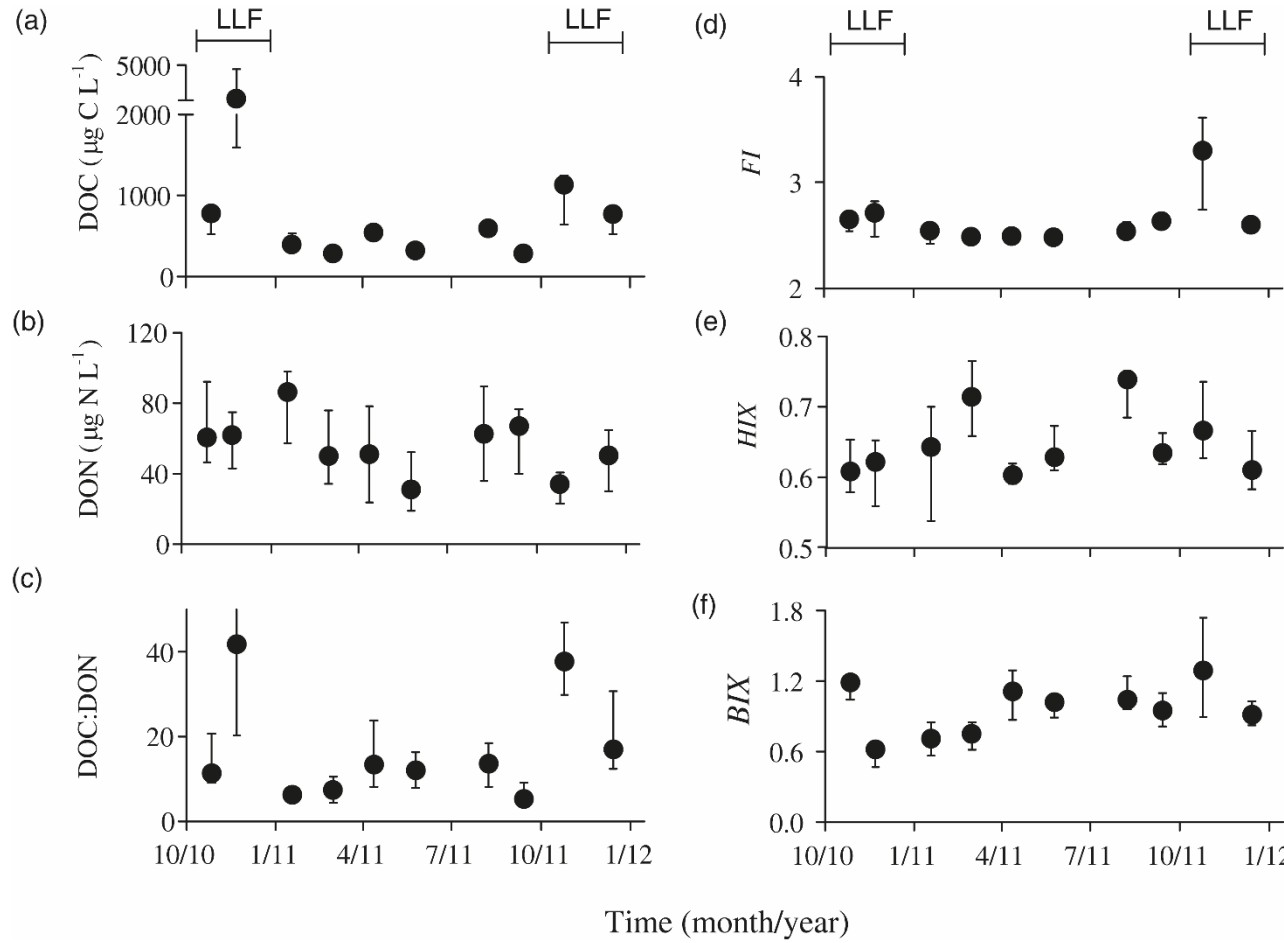

**Figure 3. Temporal pattern of (a) dissolved organic carbon (DOC), (b) dissolved organic nitrogen (DON), (c) DOC:DON molar ratio, (d) fluorescence index (*FI*), (e) humification index (*HIX*), and (f) biological index (BIX) in stream water. *FI*, *HIX*, and *BIX* were calculated from fluorescence spectroscopy. Symbols are medians and whiskers are 25th and 75th percentiles for samples collected along the main steam. The leaf litter fall period (LLF) is indicated.**

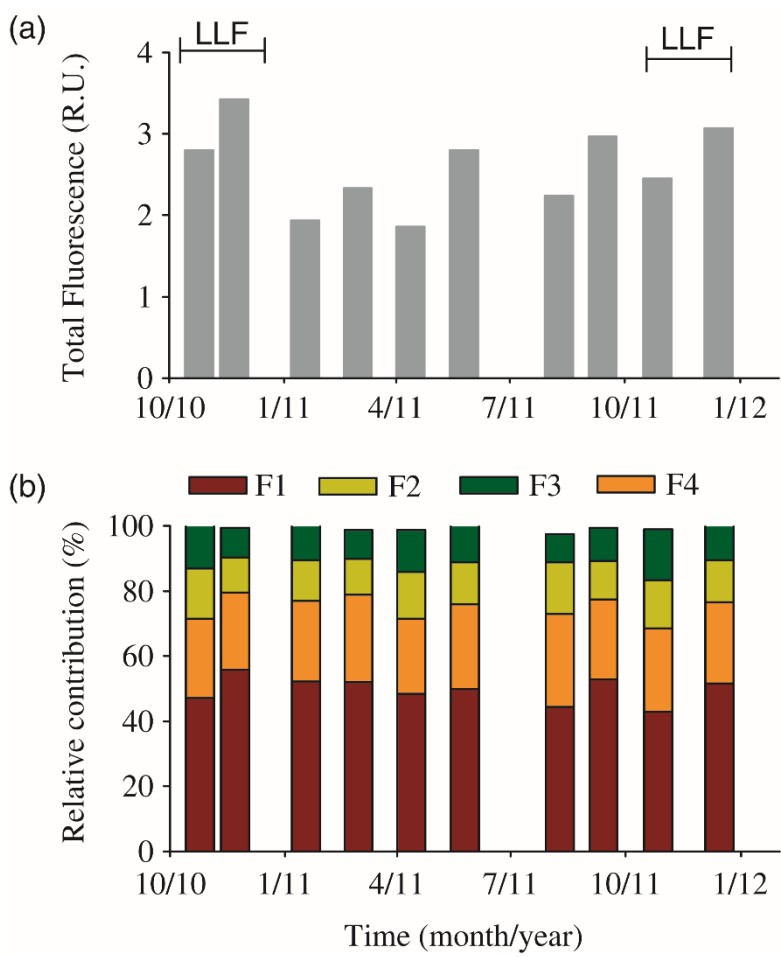

**Figure 4.** Temporal pattern of (a) total fluorescence of the four PARAFAC components and (b) their relative contribution to total fluorescence in the main steam of the Font del Regàs stream. The fluorescence components F1 and F4 corresponded to protein-like materials, while F2 and F3 corresponded to humic-like materials. Bars are median values for each sampling date. The leaf litter fall period (LLF) is indicated. R.U. are raman units. See more details on the obtained PARAFAC model in Table S1, S2, and Fig. S1 (Supplementary Information).

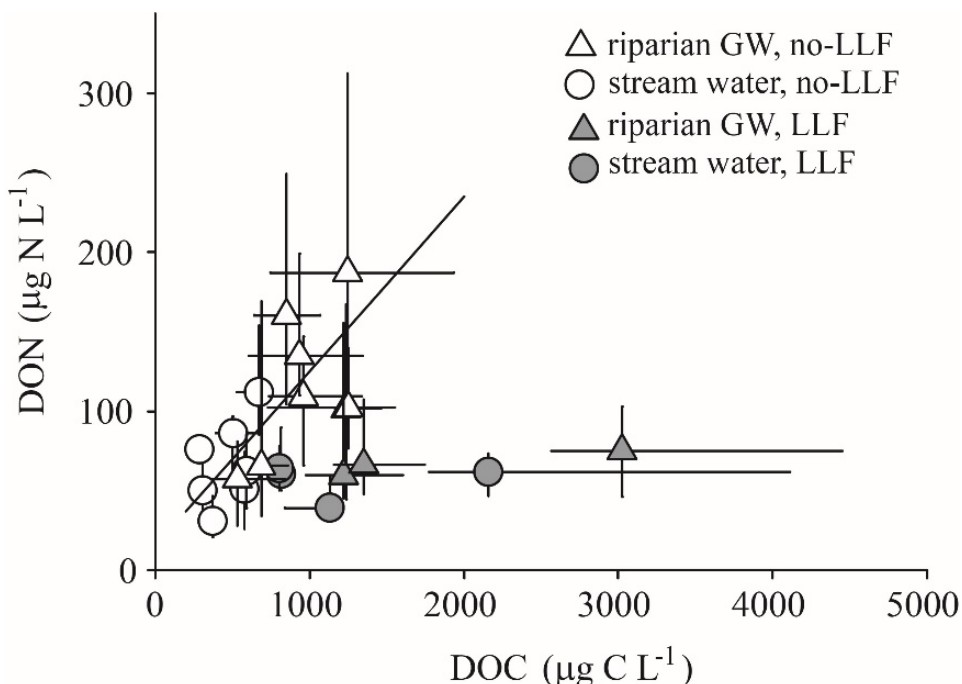

5 **Figure 5. Relationship between dissolved organic carbon (DOC) and dissolved organic nitrogen (DON) concentrations in stream water and riparian groundwater (GW). Symbols are median values and whiskers are 25th and 75th percentiles for each sampling date. The black line shows the DOC vs DON linear relationship for stream water and riparian GW samples pooled together for the non-LLF period (ANOVA, $F = 16.6$, $df = 13$, $p = 0.0015$). The relationship was not significant for the LLF period.**

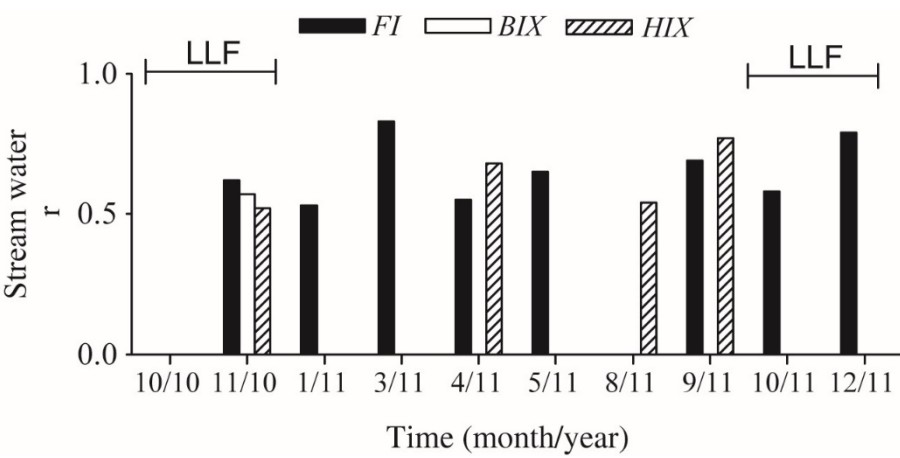

**Figure 6. Temporal pattern of the standardized regression coefficient (*r*) obtained by fitting linear regression models to values of spectroscopic indexes measured along the 4km study reach. The *r* is shown for the fluorescence index (*FI*), biological index (BIX), and humification index (HIX) in stream water. For each sampling date, r > 0 indicates that values for a particular spectroscopic index increased significantly in stream water along the study reach. Bars are shown only when the model was significant (p < 0.05).**
15 **The leaf litter fall (LLF) period is indicated. Note that none of the three spectroscopic indexes showed significant longitudinal patterns for riparian groundwater in any of the sampling dates.**

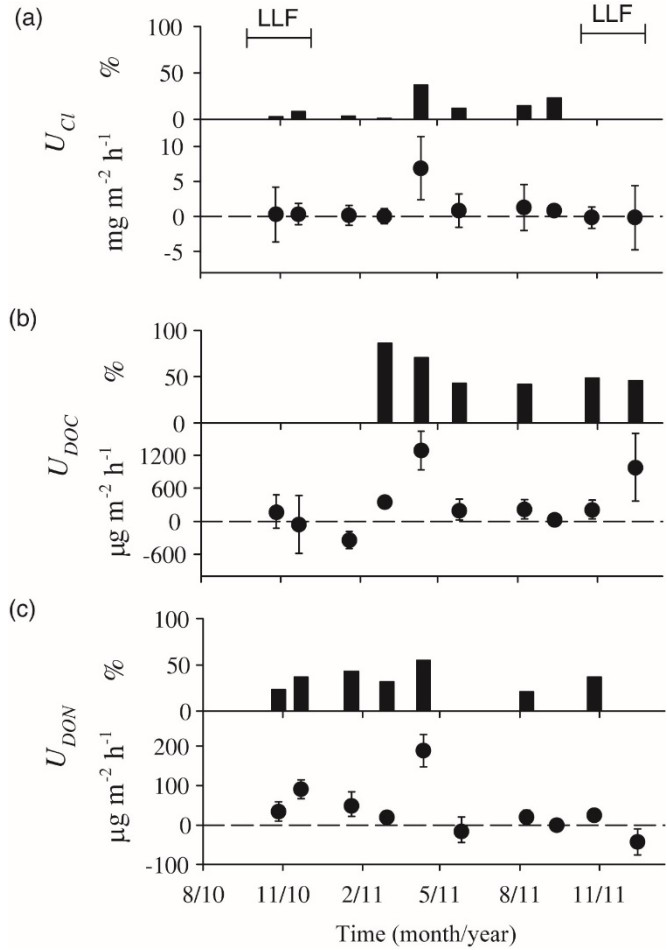

**Figure 7. Temporal pattern of in-stream net uptake ($U$, either in µg or mg m$^{-2}$ h$^{-1}$) for (a) chloride, (b) dissolved organic carbon (DOC), and (c) dissolved organic nitrogen (DON) at the whole reach scale. Whiskers are the uncertainty associated with the estimation of stream discharge from NaCl slug additions as in Bernal et al. (2015). Values of $U > 0$ indicate that gross uptake prevails over release, while $U < 0$ indicates the opposite. For cases with $U > 0$, the contribution of in-stream net uptake to decrease stream solute fluxes (i.e. $U \times A/F_{in}$, in %) is shown (black bars). The leaf litter fall period (LLF) is indicated.**

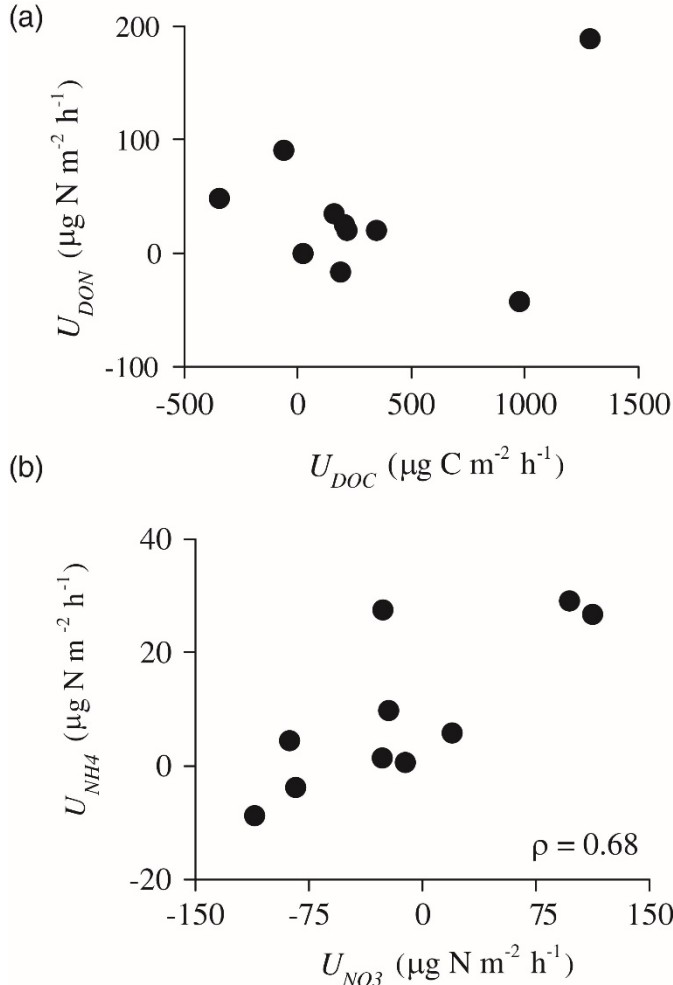

**Figure 8. Relationship between in-stream net uptake along the study reach for (a) $U_{DOC}$ and $U_{DON}$, and (b) $U_{NO3}$ and $U_{NH4}$. The Spearman coefficient ($\rho$) is shown only when significant ($p < 0.05$).**

**Tables**

Table 1. Characterization of chloride ($Cl^-$), and dissolved organic matter (DOM) (both concentrations and quality) in the main strem and in riparian groundwater (riparian GW) for the leaf litter fall period (LLF) and the non leaf litter fall period (non-LLF) at Font del Regàs. Values are medians and interquartile ranges [25th, 75th percentiles] for dissolved organic carbon (DOC) and dissolved organic nitrogen (DON) concentrations, DOC:DON molar
5   ratio, fluorescence index (*FI*), humification index (*HIX*), biological index (*BIX*), and the four PARAFAC components (F1, F2, F3, and F4). The number of cases is shown in parenthesis.

| | LLF | | | non-LLF | | |
| --- | --- | --- | --- | --- | --- | --- |
| | Stream | Riparian GW | p-value | Stream | Riparian GW | p-value |
| $Cl^-$ (mg/L) | 8.6 [7.8, 13.1] (59) | 9.8 [7.8, 13.7] (58) | 0.2 | 7.8 [7.3, 8.8] (101) | 8.7 [7.4, 10.6] (96) | 0.0174 |
| DOC (µgC/L) | 843 [643, 1243] (59) | 1411 [1133, 2311] (56) | <0.0001 | 406 [304, 580] (102) | 864 [626, 1414] (93) | <0.0001 |
| DON (µgN/L) | 48 [34, 67] (47) | 67 [45, 157] (38) | 0.012 | 63 [36, 87] (97) | 113 [64, 195] (82) | <0.0001 |
| DOC:DON | 22 [14, 43] (47) | 27 [14, 43] (38) | 0.8 | 8 [5, 15] (93) | 10 [6, 14] (82) | 0.3 |
| *Chromophoric indexes* | | | | | | |
| FI | 2.79 [2.56, 2.83] (55) | 2.59 [2.44, 2.62] (54) | 0.0001 | 2.54 [2.47, 2.59] (84) | 2.53 [2.41, 2.60] (79) | 0,211 |
| BIX | 0.60 [0.60, 0.67] (55) | 0.70 [0.63, 0.75] (54) | 0,0072 | 0.67 [0.61, 0.71] (84) | 0.67 [0.60, 0.73] (79) | 0,646 |
| HIX | 1.03 [0.66, 1.24] (55) | 1.51 [0.84, 1.82] (54) | 0.0066 | 0.94 [0.75, 1.09] (84) | 1.36 [0.86, 1.63] (79) | <0.0001 |
| *PARAFAC components* | | | | | | |
| F1 | 1.78 [1.19, 1.87] (55) | 1.70 [1.14, 1.90] (54) | 0.831 | 1.24 [0.99, 1.41] (84) | 1.31 [1.03, 1.54] (79) | 0.373 |
| F2 | 0.45 [0.32, 0.50] (55) | 0.80 [0.40, 0.94] (54) | <0.0001 | 0.31 [0.27, 0.36] (84) | 0.58 [0.36, 0.67] (79) | <0.0001 |
| F3 | 0.44 [0.28, 0.61] (55) | 0.68 [0.28, 0.79] (54) | 0.115 | 0.25 [0.20, 0.29] (84) | 0.42 [0.24, 0.47] (79) | <0.0001 |
| F4 | 0.89 [0.64, 1.02] (55) | 1.02 [0.66, 1.16] (54) | 0.021 | 0.65 [0.51, 0.77] (84) | 0.83 [0.61, 0.93] (79) | <0.0001 |

*The *p* value of the Wilcoxon paired rank sum test is shown in each case.

**Table 2. Median and interquartile range [25th, 75th] of the relative contribution of inputs from upstream ($Q_{top}$ x $C_{top}$/$F_{in}$), tributaries ($Q_{tr}$ x $C_{tr}$/$F_{in}$), net riparian groundwater ([$Q_{gw}$ x $C_{gw}$ > 0]/$F_{in}$), and in-stream release ([$Q_{sw}$ x $C_{sw}$ > 0]/$F_{in}$) to stream solute fluxes at the whole-reach scale. Note that relative contributions from different sources do not add to 100% because they are medians rather than means.**

| Relative contribution (%) | Cl⁻ | DOC | DON |
|---|---|---|---|
| *Upstream* | 15 [12, 17] | 9 [8, 13] | 52 [40, 60] |
| *Riparian groundwater* | 28 [14, 38] | 58 [41, 65] | 30 [15, 43] |
| *Tributaries* | 59 [46, 69] | 30 [17, 36] | 10 [8, 30] |
| *In-stream release* | 0 [0, 0.3] | 0 [0, 5] | 0 [0, 4] |