# Peer review of "Decoupling of dissolved organic matter patterns between stream and riparian groundwater in a headwater forested catchment"

_Hydrology and Earth System Sciences, 2017_

## Referee Comment (RC1) · Anonymous Referee #1 · 19 Oct 2017

The manuscript entitled 'Decoupling of dissolved organic matter patterns between stream and riparian groundwater in a headwater forested catchment' by Bernal and others focuses on the role of in-stream transformative processes on DOM concentration and composition by comparing the DOM found in riparian groundwater (source of DOM to the stream) and stream water across 1+ years. This type of research is key in understanding how streams potentially process and transform terrestrial DOM, which adds to our growing knowledge of streams acting as both pipes and reactors of terrestrial organic matter. The authors use a combination of approaches from calculating reach-scale DOC and DON budgets to estimate the loss or export of DOM along with the compositional characteristics of DOM (e.g. PARAFAC). The authors were able to

illustrate that streams do indeed transform and process DOM during base flow conditions in terms of concentration, but also composition. Overall, the paper was well written and I liked the approaches the authors used to address the role of in-stream transformation of terrestrial DOM. However, I have a few general and specific comments to help improve the manuscript regarding data description and interpretation.

General Comments

The tributaries contribute a significant proportion of the stream water discharge to this study reach (e.g. approximately the same stream discharge as the top of the reach, Table S3), yet there is little to no discussion of the contribution of this source to the stream. How does the influence of the tributary perhaps drive U of DOC and DON? Was the DOM composition from the tributaries similar to that of the main stem? I don't think any new analysis is needed, but simply a description of the findings and perhaps some discussion on how these tributary inputs may (or may not) drive the changes in DOM that is observed along the main stem.

Given the data set, I was missing the spatial patterns of the DOC and DON along the stream reach. It would be nice to see or give a description of the longitudinal trend of DOC and DON along the study reach. Did the 15 sampling locations along the reach very greatly in terms of DOM concentration or composition? How much did the groundwater differ along the reach? Did the Cl- concentrations, as the non-reactive anion, vary along the reach as the stream water discharge increased? How did this change in relation to the DOM? Similar to my comment above, I don't think new analysis is warranted – but simply a figure depicting an example of the potential variability of the DOM concentration (DOC and DON) and composition along this 3+ km study reach.

Referring to Figure 6, given that the error bars for UDOC and UDON overlap 0, I count 2/10 days where U>0 for DOC and 4, maybe 5 dates where U>0 for DON. I understand the median value is above 0, but given the variability (i.e. the error bars), U = 0 cannot be discounted. I suggest the authors re-cast the results to explain this result and

therefore their interpretation more clearly (in reference to text P9 L11).

Also, did the authors at all consider residence time of the reach in terms of DOM processing (e.g. Casas-Ruiz and others 2017 L&O)? What was the average velocity of the stream reach among sampling dates? And residence time would be a more important factor at base flow than storm flow. While obvious to some, perhaps mention this within the discussion.

Overall, the manuscript presentation is clear and concise. Any comments that I may have had regarding grammar or writing was extremely minor (see below).

Specific Comments

P1 L23 – non-LLF rather than no-LLF

P1 L24 – I suggest changing 'reflex' to 'reflection' (or reflects) here and throughout the manuscript. Reflex could indicate an opposite or opposing outcome whereas 'reflect' and 'reflection' indicates similarity. Similarity is what I think the authors intend within this context.

P2 L18 – This is very minor, but I would re-cast 'important' as your readers may not understand what is deemed as important in this context. Perhaps recast to 'a significant fraction'?

P3 L6 – Is there a citation for the carbon vehicle hypothesis?

P6 L7 – The authors calculated uncertainty in U (uptake or generation) based on the variability on stream water Q. Did the authors also consider the variability in the DOM and DON concentrations as well, as flux estimates will vary based both on Q and concentration variability?

P6 L11 – for consistency and clarity, instead of referring to 'release' (U < 0) as 'opposite' of uptake – call it 'release'.

P6 L17 – typo – 'steam' to 'stream'
P9 L8 – typo - 'no' to 'not' statistically significant

P9 L23 – I suggest re-cast for this latter part of the sentence — minor changes – but perhaps … 'stream water and riparian GW to investigate whether stream DOM reflected terrestrial sources or if in-stream processes modified DOM quality.' Elucidate has the same meaning as 'estimate' – but I don't think 'estimate' is the proper word here within the context of this sentence (unless you modify the sentence to 'estimate fluxes' or something similar). Also, see my comment above – 'reflex' should be 'reflect' – and were able to modify – can be simplified to 'modified'.

P10, L20 – The authors already include a number of citations explaining why the uptake rates of DOC and DON were 10-1000 fold lower than rates of in-stream DOM uptake from reported experiments, but I think they should include Mineau and others 2016 – as this particular review paper discusses that ambient DOM uptake «« than DOM uptake using simple sugars, etc...

Mineau, M. M., Wollheim, W. M., Buffam, I., Findlay, S. E. G., Hall, R. O., Jr., Hotchkiss, E. R., et al. (2016). Dissolved organic carbon uptake in streams: A review and assessment of reach-scale measurements. Journal of Geophysical Research: Biogeosciences, 121(8), 2019–2029. http://doi.org/10.1002/2015JG003204

P11 L2 – typo 'tan' to 'than'

P11 L8 – change 'what' to 'which' reinforces their potential...

P11 L9 – suggest deleting 'as merely'

---

## Referee Comment (RC2) · Anonymous Referee #2 · 30 Oct 2017

This study investigated the differences and fate of riparian groundwater and in-stream DOC and DON. The study found in-stream production and transformations of DOC that support the assertion that stream corridors serve a key role in biogeochemical cycling of carbon, beyond use as a conduit.

The study was well designed and well written. My primary concern was with the lack of context surrounding prior research into hyporheic biogeochemical cycling and the framing of groundwater within the text. The methods describe sampling "riparian groundwater" from a shallow depth near the stream edge. I am not familiar with the term "riparian groundwater," but this sounds like sampling the hyporheic zone of the stream and is

quite different from sampling pure groundwater. In addition to context being added to the introduction, I think this distinction needs to be fleshed out in the discussion.

Additional comments:

Consider consistently using "allochthonous" and "autochthonous" to reduce some of the wordiness of describing terrestrial vs. in-stream DOM.

The conclusions could benefit from describing directions for future research.

P 12 L21: Change "modify" to "modifies"

Figure 1: I would suggest finding a way to more clearly differentiate between "evergreen oak" and "other." They look quite similar in the key.

Figure 2, 3, 5, 6: You use the same x-axis notation of month/year for all of these plots, but you only list "Time (month/year)" on some of them. I was initially confused by the notation. I suggest adding "Time (month/year)" to the plots that lack it.

---

## Author Comment (AC1) · 22 Dec 2017

Center of Advanced Studies of Blanes (CEAB-CSIC)

C/Accés Cala St. Francesc 14, 17300

Blanes, Girona, Spain

Blanes, 22nd of 2017

Dear editor and reviewers of *Hydrology and Earth System Sciences*,

Please find enclosed our responses to the comments from the two referees regarding the paper "*Decoupling of dissolved organic matter patterns between stream and riparian groundwater in a headwater forested catchments*" (hess-2017-511). Overall, we are happy that you find the study interesting and a potential contribution to *Hydrol. Earth. Sys. Sci.* journal. We have thoughtfully read your helpful comments, and we are now working on the manuscript to incorporate all those changes. Following suggestions from reviewer #2, we do now consider more explicitly stream hydrology by including additional data on stream water velocity and water residence time. Moreover, we show chloride (hydrological tracer) and DOM concentrations measured in tributaries and along the 3.7km reach. Finally, we have incorporated additional explanations and discussion regarding the uncertainties associated with the mass balance approach and the difficulty of delimiting water compartments at the stream-hyporheic-riparian interface, as highlighted by reviewer #3.

Below you will find detailed responses to each of the general comments as well as to the most substantial specific comments made by the reviewers Overall, we believe that we can successfully solve all the points raised by the reviewers, and generate an improved version of the manuscript including their suggestions.

Please, do not hesitate to contact us if further clarifications are needed at this stage.

Sincerely,

Susana Bernal

Cc: Anna Lupon, Núria Catalan, Sara Castelar, and Eugènia Martí.

**Anonymous Referee #2**

**REFEREE:** *The manuscript entitled 'Decoupling of dissolved organic matter patterns between stream and riparian groundwater in a headwater forested catchment' by Bernal and others focuses on the role of in-stream transformative processes on DOM concentration and composition by comparing the DOM found in riparian groundwater (source of DOM to the stream) and stream water across 1+ years. This type of research is key in understanding how streams potentially process and transform terrestrial DOM, which adds to our growing knowledge of streams acting as both pipes and reactors of terrestrial organic matter. The authors use a combination of approaches from calculating reach-scale DOC and DON budgets to estimate the loss or export of DOM along with the compositional characteristics of DOM (e.g. PARAFAC). The authors were able to illustrate that streams do indeed transform and process DOM during base flow conditions in terms of concentration, but also composition. Overall, the paper was well written and I liked the approaches the authors used to address the role of in-stream transformation of terrestrial DOM. However, I have a few general and specific comments to help improve the manuscript regarding data description and interpretation.*

**AUTHORS:** Many thanks for your positive comments. We are glad that you enjoyed the paper and that you consider that "*this type of research is key*" in understanding the cycling of DOM in stream ecosystems. We have carefully considered all your suggestions and we are working to incorporate them in the new version of the manuscript.

**General Comments**

**REFEREE:** *The tributaries contribute a significant proportion of the stream water discharge to this study reach (e.g. approximately the same stream discharge as the top of the reach, Table S3), yet there is little to no discussion of the contribution of this source to the stream. How does the influence of the tributary perhaps drive U of DOC and DON? Was the DOM composition from the tributaries similar to that of the main stem? I don't think any new analysis is needed, but simply a description of the findings and perhaps some discussion on how these tributary inputs may (or may not) drive the changes in DOM that is observed along the main stem.*

**AUTHORS:** That's right, thank you for rising up this point. In a previous study conducted in the same reach, we showed that permanent tributaries comprise about 50% of the catchment area and contribute by 56% to stream discharge (Bernal et al., 2015). This information is now included in the Study Site section. Thus, as you suggest, one would expect tributaries to have a strong influence on determining DOM fluxes.

To explore this question, we have calculated the contribution of the different water sources (upstream, tributaries, riparian groundwater, and in-stream release) to chloride (Cl⁻) and DOM stream input fluxes (Table R1). The results showed that the contribution of tributaries to stream DOM fluxes was relatively small (from 10 to 30%) compared to stream Cl⁻ fluxes (>50%), suggesting that other sources of DOM within the catchment were more important than tributaries. For instance, riparian groundwater was the most important source of DOC along the reach, while upstream sources provide most of the DON. These differences between DOM sources could be explained by differences in vegetation and topography: the upstream site drained a north facing catchment mostly covered with beech, while the two main tributaries drained a southeast facing catchment mostly covered with oak (Figure 1, main manuscript). Therefore, it is also possible that there could be differences in DOM quality between tributaries and other inputs sources. Unfortunately, we did not measure DOM quality

in the tributaries, but we can (and will) rise this point in the discussion. Moreover, we will incorporate how input fluxes were calculated as well as Table R1 and associated results and discussion in the revised version of the manuscript.

In addition, we will refer more clearly to chloride and DOM concentrations measured in the tributaries by including their water chemistry in new Figure 2 as well as in the main text of the results section (please, see our detailed responses to the next general comment).

**Table R1.** Median and interquartile range [25th, 75th] of the relative contribution of inputs from upstream ($Q_{top} x C_{top}/F_{in}$), tributaries ($Q_{tr} x C_{tr}/F_{in}$), net riparian groundwater ([$Q_{gw} x C_{gw} > 0]/F_{in}$), and in-stream release ([$Q_{sw} x C_{sw} > 0]/F_{in}$) to stream solute fluxes at the whole-reach scale. Note that relative contributions from different sources do not add to 100% because they are medians rather than means.

| Relative contribution (%) | Cl- | DOC | DON |
|---|---|---|---|
| Upstream | 15 [12, 17] | 9 [8, 13] | 52 [40, 60] |
| Riparian groundwater | 28 [14, 38] | 58 [41, 65] | 30 [15, 43] |
| Tributaries | 59 [46, 69] | 30 [17, 36] | 10 [8, 30] |
| In-stream release | 0 [0, 0.3] | 0 [0, 5] | 0 [0, 4] |

**REFEREE:** *Given the data set, I was missing the spatial patterns of the DOC and DON along the stream reach. It would be nice to see or give a description of the longitudinal trend of DOC and DON along the study reach. Did the 15 sampling locations along the reach very greatly in terms of DOM concentration or composition? How much did the groundwater differ along the reach? Did the Cl- concentrations, as the non-reactive anion, vary along the reach as the stream water discharge increased? How did this change in relation to the DOM? Similar to my comment above, I don't think new analysis is warranted – but simply a figure depicting an example of the potential variability of the DOM concentration (DOC and DON) and composition along this 3+ km study reach.*

**AUTHORS:** Following your suggestion, we have prepared a new figure that will be included in the main manuscript. The figure shows median and percentile concentrations for chloride, DOC, and DON along the reach for both the LLF and non-LLF period (Figure R1). Moreover, statistical significant longitudinal trends in concentration have been indicated with solid lines, and the relative increase in concentration along the reach will be indicated in the text of the results sections (for significant longitudinal trends). We believe that this figure and associated results are helpful to describe trends for chloride and DOM along the reach and, further, provides information on how variable stream water concentrations were along the reach. Note that this figure also includes median stream water concentration from tributaries.

The following sentences related to the new Figure 3 will be included in the Results section 4.1 (underlined text):

"During the study period, median Cl- concentration in the main stream was higher for the LLF (8.6 [7.8, 13.1] [25th, 75th percentiles] mg L-1) than for the non-LLF period (7.8 [7.3, 8.8] mg L-1) (Mann Whitney test, Z =2.82, df = 1, p = 0.005). Stream Cl- concentrations increased by 43% and 48 % during the LLF and the non-LLF period, respectively (Figure 2a). In the tributaries, median stream Cl- concentration was 10.2 [8.8, 14.2] mg L-1. For DOC, median DOC concentration in the main stream was higher for the LLF (843 [643, 1243] μg C L-1) than for the non-LLF period (406 [304, 580] μg C L-1) (Mann Whitney test, Z =2.55, df = 1, p = 0.008) (Fig. 3a). Stream DOC concentrations increased along the reach by 58%

during the LLF period (Figure 2b). In the tributaries, median stream DOC concentration was 577 [390, 881] μg C L⁻¹. For DON, median stream DON concentration in the main stream was 58 [35, 78] μg N L⁻¹ and showed no seasonal pattern (Mann Whitney test, Z = -0.85, df = 1, p > 0.05) (Fig. 2b3b). Stream DON concentrations showed no clear longitudinal changes in any of the two study periods (Fig. 2c), though concentrations could vary by 40% on a single date. In the tributaries, median stream DON concentration was 54 [34, 75] μg N L⁻¹. The median stream DOC:DON ratio in the main stream was higher during the LLF (DOC:DON = 22 [14, 43]) than during the non-LLF period (DOC:DON = 8 [5, 15]) (Mann Whitney test, Z = 1.98, df = 1, p = 0.033) (Fig. 3c)."

[Figure]

**Figure R1 (will be new Figure 2).** Longitudinal patterns of (a) chloride, (b) dissolved organic carbon (DOC), and (c) dissolved organic nitrogen (DON) concentrations in stream water along the 3.7km reach. Symbols are median values and whiskers are the interquartile range (25th, 75th percentiles) for the main stream (circles) and tributaries (diamonds). Concentrations are shown separately for the LLF (grey) and non-LLF period (white). Black circles in (b) correspond to the field campaign of the 22 of November of 2010 when DOC concentrations were higher than for the remaining study period. Model regressions are indicated with solid lines only when significant (tributaries not included in the model).

Moreover, we have built a similar figure for riparian groundwater so that the reader can evaluate by how much chloride and DOM concentration in riparian groundwater varied along the reach (Figure R2). We will refer to this Figure in the Results section, but we have

decided to include it as Supplementary Material in order to keep the paper as concise as possible and avoid losing focus, which is the stream compartment and in-stream processes.

[Figure]

**Figure R2 (will be new Figure S3).** Longitudinal patterns of (a) chloride, (b) dissolved organic carbon (DOC), and (c) dissolved organic nitrogen (DON) concentrations in riparian groundwater along the reach. Symbols are median values and whiskers are the interquartile range (25th, 75th percentiles). Concentrations are shown separately for the LLF (grey) and non-LLF period (white).

Regarding the variability of DOM composition along the reach, we would like to highlight that we already provided information in this regard in the former version of the manuscript. In particular, Figure 5 showed the temporal pattern of the standardized regression coefficient obtained by fitting a linear regression to the values of FI, BIX, and HIX along the reach. Moreover, an additional figure was included in the Supplementary materials showing the same type of calculation for the PARAFAC components (Figure S2, former version of the manuscript).

Regarding spatiotemporal changes in stream discharge and chloride concentration, note that most of the information requested was included in a previous paper focused on stream nutrient dynamics along the same study reach (Bernal et al. 2015). To avoid redundancies, we first decided to not repeat these results in this paper. Yet, we agree with the

reviewer that some additional information about hydrology could be helpful for contextualizing our results. In Bernal et al. (2015), we found that despite stream discharge increased along the reach, area-specific discharge decreased longitudinally. These results indicate that hydrological retention increased at the valley bottom compared to upstream segments. The following sentence will be included in the Study Site section (text underlined):

> "Mean stream discharge increased along the reach from 20 to 70 L s$^{-1}$. During the study period, the stream gained water in net terms along the reach but it lost water towards the riparian zone, especially during the summer months. Moreover, mean area-specific stream discharge decreased longitudinally, an indication that on average hydrological retention was higher at the valley bottom compared to upstream segments (Bernal et al. 2015)".

Additional information of hydrology will be incorporated in the text by including patterns in Cl$^-$ concentrations. Specifically:
  (1) Cl$^-$ concentrations in stream water and riparian groundwater for the two periods of study will be included in Table 1.
  (2) longitudinal trends in Cl$^-$ concentration along the reach for the two periods will be shown in new Figure 2 (this is Figure R1).

New results associated to Table 1 and Figure R1 will be included in section 4.1, 4.2., and 4.3, and referred in the discussion when appropriated.

**REFEREE:** *Referring to Figure 6, given that the error bars for UDOC and UDON overlap 0, I count 2/10 days where U>0 for DOC and 4, maybe 5 dates where U>0 for DON. I understand the median value is above 0, but given the variability (i.e. the error bars), U = 0 cannot be discounted. I suggest the authors re-cast the results to explain this result and C2 therefore their interpretation more clearly (in reference to text P9 L11).*

**AUTHORS:** We are glad that you have highlighted this issue. Thanks to your comment, we have realized that error bars associated with $U_{DOC}$ values were so large and asymmetric compared to $U_{DON}$ and $U_{Cl}$ because of a mistake when calculating min and max fluxes from the tributaries. After correcting this mistake, former Figure 6 looks as follows:

[Figure]

**Figure 6 (new Figure 7).** Temporal pattern of in-stream net uptake ($U$, either in µg or mg m$^{-2}$ h$^{-1}$) for (a) chloride, (b) dissolved organic carbon (DOC), and (c) dissolved organic nitrogen (DON) at the whole reach scale. Whiskers are the uncertainty associated with the estimation of stream discharge from NaCl slug additions as in Bernal et al. (2015). Values of $U > 0$ indicate that gross uptake prevails over release, while $U < 0$ indicates the opposite. For cases with $U > 0$, the contribution of in-stream net uptake to decrease stream solute fluxes (i.e. $U \times A/F_{in}$, in %) is shown (black bars). The leaf litter fall period (LLF) is indicated.

Following your suggestion, we have recalculated the number of cases for which the stream was acting as a net sink of DOM considering only those cases for which $U > 0$ for both median and error bars. To increase the clarity of our response, we include values of $U_{Cl}$, $U_{DOC}$, and $U_{DON}$ in Table R2. The results showed that the stream acts as a net sink of DOM ($U > 0$) in six (rather than in seven) and in seven (rather than in eight) out of 10 sampling dates for DOC and DON, respectively. In these cases, in-stream processes contributed to reduce stream fluxes by 47 [43, 65] % (rather than by 44 [36, 54] %) and 37 [28, 40] % for DOC and DON, respectively. We have already corrected these results in section 4.4 (former P9 L11).

**Table R2.** In-stream net uptake rates ($U$, either in µg or mg $m^{-2}$ $h^{-1}$) for chloride, dissolved organic carbon (DOC), and dissolved organic nitrogen (DON) at the whole reach scale. The upper and lower limit of $U$ are based on the empirical uncertainty associated with discharge measurements as in Bernal et al. (2015). Values of $U > 0$ (in blue) indicate that gross uptake prevails over release, while $U < 0$ (in red) indicates the opposite.

| | $U_{Cl}$ (mg $m^{-2}$ $h^{-1}$) | | | $U_{DOC}$ ($\mu g$ C $m^{-2}$ $h^{-1}$) | | | $U_{DON}$ ($\mu g$ N $m^{-2}$ $h^{-1}$) | | |
|---|---|---|---|---|---|---|---|---|---|
| 27/10/2010 | -3,7 | 0,3 | 4,2 | -132,4 | 162,1 | 480,0 | 9,5 | 34,2 | 58,4 |
| 22/11/2010 | -1,2 | 0,3 | 1,9 | -587,0 | -61,0 | 469,9 | 66,2 | 90,6 | 114,3 |
| 19/01/2011 | -1,3 | 0,1 | 1,5 | -493,3 | -344,7 | -181,8 | 21,5 | 47,7 | 83,5 |
| 01/03/2011 | -1,0 | 0,0 | 1,1 | 265,4 | 348,0 | 433,8 | 9,1 | 19,5 | 30,4 |
| 12/04/2011 | 2,4 | 6,9 | 11,4 | 939,8 | 1287,5 | 1635,2 | 147,5 | 188,5 | 229,4 |
| 26/05/2011 | -1,5 | 0,8 | 3,2 | 24,9 | 189,7 | 403,1 | -43,8 | -17,0 | 20,7 |
| 09/08/2011 | -2,0 | 1,2 | 4,5 | 49,0 | 216,1 | 393,5 | 3,1 | 19,8 | 38,1 |
| 13/09/2011 | 0,1 | 0,8 | 1,6 | -2,9 | 23,7 | 51,4 | -4,5 | -0,3 | 3,7 |
| 26/10/2011 | -1,7 | -0,2 | 1,4 | 39,3 | 205,7 | 383,3 | 18,1 | 24,8 | 32,1 |
| 15/12/2011 | -4,8 | -0,2 | 4,4 | 364,5 | 976,5 | 1596,5 | -75,7 | -42,8 | -9,9 |

**REFEREE:** *Also, did the authors at all consider residence time of the reach in terms of DOM processing (e.g. Casas-Ruiz and others 2017 L&O)? What was the average velocity of the stream reach among sampling dates? And residence time would be a more important factor at base flow than storm flow. While obvious to some, perhaps mention this within the discussion.*

**AUTHORS:** As stated in the former M&M and Discussion, sampling campaigns were conducted during base flow conditions. Thus, the variability of hydrological conditions among sampling dates was not large enough to explore the influence of hydrology on in-stream DOM processing. For example, mean water velocity range from 0.23 to 0.4 m $s^{-1}$ (mean values now included in the Study Site section), while water residence time within the reach ranged from 2.3 to 4.9 h. These values fall within the low range of values reported by Casas-Ruiz et al. (2017) (water residence time from 0.5 to 100 h), where many stream reaches included, or were under the influence of, small reservoirs. Therefore, our data is representative of base flow conditions in running waters of headwater catchments that do not contain natural or artificial water storage structures. Following the reviewer suggestion, we have modified the following paragraph as follows (text underlined) (P11, L5):

> "Yet, our results are representative of base flow conditions which represent ca. 60 % of the annual DOC and DON flux in the study catchment (unpublished data). Moreover, mean water residence time was relatively low (4h, unpublished data) because running waters predominated and there were no natural or artificial dams along the study reach. Further studies including storm flow conditions and/or reaches with small reservoirs would be needed to gain a more complete picture of the role of in-stream processes on DOM dynamics and whether headwater streams shift from reactors to pipes with changing hydrological conditions (e.g., Casas-Ruíz et al., 2017; Raymond et al., 2016)".

**REFEREE:** *Overall, the manuscript presentation is clear and concise. Any comments that I may have had regarding grammar or writing was extremely minor (see below).*

**AUTHORS:** Many thanks for your positive comments and grammar suggestions.

**Specific Comments**

**AUTHORS:** All the small specific comments have been incorporated in the main text of the revised version of the manuscript. Mineau et al. (2016) has been included as a reference in the Discussion. Thanks for the language suggestions, they are helpful and we have learned from your explanations. For those specific comments that were more substantial, we include a brief answer below:

P1 L23 – *non-LLF rather than no-LLF*. **OK**

P1 L24 – *I suggest changing 'reflex' to 'reflection' (or reflects) here and throughout the manuscript. Reflex could indicate an opposite or opposing outcome whereas 'reflect' and 'reflection' indicates similarity. Similarity is what I think the authors intend within this context*. **OK**

P2 L18 – *This is very minor, but I would re-cast 'important' as your readers may not understand what is deemed as important in this context. Perhaps recast to 'a significant fraction'?*. **OK**

P3 L6 – *Is there a citation for the carbon vehicle hypothesis?* **OK**

P6 L7 – *The authors calculated uncertainty in U (uptake or generation) based on the variability on stream water Q. Did the authors also consider the variability in the DOM and DON concentrations as well, as flux estimates will vary based both on Q and concentration variability?*

**AUTHORS:** Yes, the reviewer is right: the uncertainty of *U* was based on the error associated with the measurement of stream discharge. Unfortunately, we only collected one stream water sample (from the thalweg) from each sampling site on each field campaign. Thus, we could not consider the empirical error associated with stream water chemistry. Following your suggestion, we will add a sentence in the first paragraph of the Discussion highlighting the uncertainties associated to the mass balance approach.

P6 L11 – *for consistency and clarity, instead of referring to 'release' (U < 0) as 'opposite' of uptake – call it 'release'.* **OK**

P6 L17 – *typo – 'steam' to 'stream'.* **OK**

P9 L8 – *typo - 'no' to 'not' statistically significant.* **OK**

P9 L23 – *I suggest re-cast for this latter part of the sentence — minor changes – but perhaps . . . 'stream water and riparian GW to investigate whether stream DOM reflected terrestrial sources or if in-stream processes modified DOM quality.' Elucidate has the same meaning as 'estimate' – but I don't think 'estimate' is the proper word here within the context of this sentence (unless you modify the sentence to 'estimate fluxes' or something similar). Also, see my comment above – 'reflex' should be 'reflect' – and were able to modify – can be simplified to 'modified'.* **OK**

P10, L20 – *The authors already include a number of citations explaining why the uptake rates of DOC and DON were 10-1000 fold lower than rates of in-stream DOM uptake from reported experiments, but I think they should include Mineau and others 2016 – as this particular review paper discusses that ambient DOM uptake «« than DOM uptake using simple sugars, etc. Mineau, M. M., Wollheim, W. M., Buffam, I., Findlay, S. E. G., Hall, R. O., Jr., Hotchkiss, E. R., et al. (2016). Dissolved organic carbon uptake in streams: A review and assessment of reach-scale measurements. Journal of Geophysical Research: Biogeosciences, 121(8), 2019–2029. http://doi.org/10.1002/2015JG003204.* **OK**

P11 L2 – *typo 'tan' to 'than'.* **OK**

P11 L8 – *change 'what' to 'which' reinforces their….* **OK**

**Anonymous Referee #3**

**REFEREE:** *This study investigated the differences and fate of riparian groundwater and in-stream DOC and DON. The study found in-stream production and transformations of DOC that support the assertion that stream corridors serve a key role in biogeochemical cycling of carbon, beyond use as a conduit. The study was well designed and well written. My primary concern was with the lack of context surrounding prior research into hyporheic biogeochemical cycling and the framing of groundwater within the text. The methods describe sampling "riparian groundwater" from a shallow depth near the stream edge. I am not familiar with the term "riparian groundwater," but this sounds like sampling the hyporheic zone of the stream and is quite different from sampling pure groundwater. In addition to context being added to the introduction, I think this distinction needs to be fleshed out in the discussion.*

**AUTHORS:** Many thanks for your positive comments. We are glad that you find "*the study well designed and well written*". Regarding your main concern, we agree that we could be more clear when defining water body compartments. This is, indeed, an essential topic in stream ecology given the difficulty of clearly defining boundaries between the stream, the hyporheic zone and the riparian groundwater.

A priori, riparian groundwater could be considered the water laterally transported from the saturated riparian zone to the stream, while the hyporheic zone would be defined as the subsurface streambed zone that receives water from both the riparian and stream compartments (Bencala et al. 1993). From a hydrological point of view, these definitions become fuzzy because stream water can eventually infiltrate towards the riparian zone. Thus, a more relaxed definition of the hyporheic zone should include the near-stream zone too, because stream water and saturated groundwater can mix along both the vertical and horizontal dimension (that is, not strictly within the streambed zone) (Bencala et al., 2011). In any case, what is clear is that riparian groundwater must traverse both near stream and streambed zones before reaching free flowing waters. From a biogeochemical point of view, groundwater DOM is likely processed along this riparian-stream interface (e.g., Fasching et al. 2015), which makes extremely difficult to identify the exact chemistry of the riparian groundwater entering into the stream (Brookshire et al., 2009).

In this study, and for practical reasons, we assumed that the so call "riparian groundwater" (groundwater collected from 1.5 to 2m from the stream channel) was representative of the chemical signature of terrestrial groundwater entering into the stream-hyporheic zone after draining both hillslope and riparian ecosystems. Yet, we cannot rule out that the stream-groundwater mixing front could eventually move towards the riparian zone. Further, we acknowledge that we cannot distinguish whether DOM biogeochemical processing occurred in the stream water column and/or within the hyporheic zone. Thus, and regarding the interpretation of the mass balance results, we agree that it is important to highlight that no distinction could be made between the hyporheic zone and the stream water column.

Following your suggestion, we will explicitly refer to the hyporheic zone in the Introduction, and clarified the definitions and assumptions regarding riparian groundwater as indicated in the earlier paragraph. Moreover, we will specify that, for all mass balance calculations, $U_{DOM}$ encompassed all processes occurring at the riparian-stream interface (including both the hyporheic zone and the stream water column).

**Additional comments:**

**REFEREE:** *Consider consistently using "allochthonous" and "autochthonous" to reduce some of the wordiness of describing terrestrial vs. in-stream DOM.*

**AUTHORS:** Thanks for your suggestion. We will consider whether the substitution that you propose could be helpful to improve the flow of the paper; we will carefully read the manuscript and make the necessary changes in this regard.

**REFEREE:** *The conclusions could benefit from describing directions for future research.*

**AUTHORS:** Following your suggestion, we will include some sentences about future directions in the conclusion section, which will be named "*Conclusion and future research*". From our point of view, further work is needed on understanding how environmental variables shape in-stream DOM processing, in particular nutrient availability and water residence time. In this sense, the use of high frequency monitoring sensors could be helpful to gain understanding on how DOM quality and DOM processing change during storm events. Furthermore, future studies aimed to better understand the different mechanism underlying DOC and DON processing within streams are also needed.

P 12 L21: *Change "modify" to "modifies"* **OK**

**REFEREE:** *Figure 1: I would suggest finding a way to more clearly differentiate between "evergreen oak" and "other." They look quite similar in the key.*

**AUTHORS:** Following your suggestion, have modified the color key of this figure to improve clarity.

[Figure]

**Figure 1 (revised)**. Map of the Font del Regàs catchment within the Montseny Natural Park (NE, Spain). The vegetation cover and the main stream sampling stations along the 3.7-km reach are indicated. Four permanent tributaries discharged to the main stream from the upstream- to the downstream-most site (white circles). The remaining tributaries were dry during the study period. The location of the hillslope springs and soil lixiviates plots is also shown.

**REFEREE:** *Figure 2, 3, 5, 6: You use the same x-axis notation of month/year for all of these plots, but you only list "Time (month/year)" on some of them. I was initially confused by the notation. I suggest adding "Time (month/year)" to the plots that lack it.*

**AUTHORS:** Many thanks. X axis read now the same in former Figures 2, 3, 5, and 6.

**REFERENCES INCLUDED IN THE RESPONSE LETTER**

Bencala, K. E.: A perspective on stream-catchment connections, J. N. Am. Benthol. Soc., 12, 44-47, 1993. doi:10.2307/1467684.

Bencala, K. E., M. N. Gooseff, and B. A. Kimball: Rethinking hyporheic flow and transient storage to advance understanding of stream-catchment connections, Water Resources Research 47, W00H03, 2011. doi:10.1029/2010WR010066.

Bernal, S., Lupon, A., Ribot, M., Sabater, F., and Martí, E.: Riparian and in-stream controls on nutrient concentrations and fluxes in a headwater forested stream. Biogeosciences 12, 1941-1954, 2015.

Brookshire, J.E., Valett, H.M., and Gerber, S.: Maintenance of terrestrial nutrient loss signatures during in-stream transport. Ecology 90, 293-299, 2009.

Casas-Ruíz, J.P., Catalán, N., Gómez-Gener, Ll., von Schiller, D., Obrador, B., Kothawala, D.L., López, P., Sabater, S., and Marcé, R.: A tale of pipes and reactors: controls on the in-stream dynamics of dissolved organic matter in rivers. Limnology and Oceanography 62, 85-89, 2017.

Fasching, C., Ulseth, A.J., Schelker, J., Steniczka, G., Battin, T.J.: Hydrology controls dissolved organic matter export and composition in an Alpine stream and its hyporheic zone. Limnology and Oceanography, 61, 558-571, 2015. http://dx.doi.org/10.1002/lno.10232

---

## Author Response (AR1)

Center of Advanced Studies of Blanes (CEAB-CSIC)

C/Accés Cala St. Francesc 14, 17300

Blanes, Girona, Spain

Blanes, 22th of January of 2018

Dear editor and reviewers of *Hydrology and Earth System Sciences*,

Please find enclosed our responses to the comments from the two referees regarding the paper "*Decoupling of dissolved organic matter patterns between stream and riparian groundwater in a headwater forested catchments*" (hess-2017-511). Overall, we are happy that you find the study interesting and a potential contribution to *Hydrol. Earth. Sys. Sci.* journal. Despite being minor, we have thoughtfully read all your comments and suggestions, and we have thoughtfully worked to incorporate all those changes in the new version of the manuscript. Following suggestions from reviewer #2, we do now consider more explicitly stream hydrology by including additional data on stream water velocity and water residence time. Moreover, we show chloride (hydrological tracer) and DOM concentrations measured in tributaries and along the 3.7km reach. Finally, we have incorporated additional explanations and discussion regarding the uncertainties associated with the mass balance approach and the difficulty of delimiting water compartments at the stream-hyporheic-riparian interface, as highlighted by reviewer #3. Overall, we believe that we have successfully solved all the points raised by the reviewers, and that the new version of the manuscript is substantially improved.

Below you will find detailed responses to each of the general comments as well as to the most substantial specific comments made by the reviewers. When appropriate, we indicate the page and line of the revised version where changes have been made to facilitate the task of the reviewers.

Please, do not hesitate to contact us if further clarifications are needed at this stage.

Sincerely,

Susana Bernal

Cc: Anna Lupon, Núria Catalan, Sara Castelar, and Eugènia Martí.

**Anonymous Referee #2**

**REFEREE:** *The manuscript entitled 'Decoupling of dissolved organic matter patterns between stream and riparian groundwater in a headwater forested catchment' by Bernal and others focuses on the role of in-stream transformative processes on DOM concentration and composition by comparing the DOM found in riparian groundwater (source of DOM to the stream) and stream water across 1+ years. This type of research is key in understanding how streams potentially process and transform terrestrial DOM, which adds to our growing knowledge of streams acting as both pipes and reactors of terrestrial organic matter. The authors use a combination of approaches from calculating reach-scale DOC and DON budgets to estimate the loss or export of DOM along with the compositional characteristics of DOM (e.g. PARAFAC). The authors were able to illustrate that streams do indeed transform and process DOM during base flow conditions in terms of concentration, but also composition. Overall, the paper was well written and I liked the approaches the authors used to address the role of in-stream transformation of terrestrial DOM. However, I have a few general and specific comments to help improve the manuscript regarding data description and interpretation.*

**AUTHORS:** Many thanks for your positive comments. We are glad that you enjoyed the paper and that you consider that "*this type of research is key*" in understanding the cycling of DOM in stream ecosystems. We have carefully considered all your suggestions and incorporated them in the new version of the manuscript.

**General Comments**
**REFEREE:** *The tributaries contribute a significant proportion of the stream water discharge to this study reach (e.g. approximately the same stream discharge as the top of the reach, Table S3), yet there is little to no discussion of the contribution of this source to the stream. How does the influence of the tributary perhaps drive U of DOC and DON? Was the DOM composition from the tributaries similar to that of the main stem? I don't think any new analysis is needed, but simply a description of the findings and perhaps some discussion on how these tributary inputs may (or may not) drive the changes in DOM that is observed along the main stem.*

**AUTHORS:** That's right, thank you for rising up this point. In a previous study, we showed that permanent tributaries comprise about 50% of the catchment area and contribute by 56% to stream discharge (Bernal et al., 2015). This information is now included in the Study Site section (P4, L2). Thus, as you suggest, one would expect tributaries to have a strong influence on determining DOM fluxes.

To explore this question, we have calculated the contribution of the different water sources (upstream, tributaries, riparian groundwater, and in-stream release) to chloride (Cl⁻) and DOM stream input fluxes (Table R1). The results showed that the contribution of tributaries to stream DOM fluxes was relatively small (from 10 to 30%) compared to stream Cl⁻ fluxes (>50%), suggesting that other sources of DOM within the catchment were more important than tributaries. For instance, riparian groundwater was the most important source of DOC along the reach, while upstream sources provide most of the DON. These differences could be partially explained by changes in vegetation: the upstream sites had no riparian zone and drained beech forests exhibiting low mineralization and nitrification rates (Lupon et al., 2016), while most of the mid- and down-stream sites along the reach were flanked by a well-developed riparian forest that hold higher soil N processing rates (Lupon et al., 2016). These explanations, have been included in the Discussion of the revised version (P12, L11-17).

Therefore, it is also possible that there could be differences in DOM quality between tributaries and other inputs sources. Unfortunately, we did not measure DOM quality in the tributaries, and thus, we have no way to figure that out.

In addition to these changes, we have explained in M&M how input fluxes were calculated (P6, L28), and Table R1 is now included in the Results section (new Table 2). We have added the following results associated with Table 2 (P9, L24-26):

> "Riparian GW was the most important source of DOC along the reach (58% of the total inputs), while upstream sources provided most of the DON to the stream (30% of the total inputs) (Table 2). The contribution of tributaries to stream DOM fluxes was relatively small compared to stream Cl⁻ fluxes (Table 2)".

Moreover, chloride and DOM concentrations measured in the tributaries are now included in new Figure 2, and these results are referred in the main text of the results section (please, see our detailed responses to the next general comment).

**Table R1.** Median and interquartile range [$25^{th}$, $75^{th}$] of the relative contribution of inputs from upstream ($Q_{top}$ x $C_{top}$/$F_{in}$), tributaries ($Q_{tr}$ x $C_{tr}$/$F_{in}$), net riparian groundwater ([$Q_{gw}$ x $C_{gw}$ > 0]/$F_{in}$), and in-stream release ([$Q_{sw}$ x $C_{sw}$> 0]/$F_{in}$) to stream solute fluxes at the whole-reach scale. Note that relative contributions from different sources do not add to 100% because they are medians rather than means.

| Relative contribution (%) | Cl- | DOC | DON |
|---|---|---|---|
| Upstream | 15 [12, 17] | 9 [8, 13] | 52 [40, 60] |
| Riparian groundwater | 28 [14, 38] | 58 [41, 65] | 30 [15, 43] |
| Tributaries | 59 [46, 69] | 30 [17, 36] | 10 [8, 30] |
| In-stream release | 0 [0, 0.3] | 0 [0, 5] | 0 [0, 4] |

**REFEREE:** *Given the data set, I was missing the spatial patterns of the DOC and DON along the stream reach. It would be nice to see or give a description of the longitudinal trend of DOC and DON along the study reach. Did the 15 sampling locations along the reach very greatly in terms of DOM concentration or composition? How much did the groundwater differ along the reach? Did the Cl- concentrations, as the non-reactive anion, vary along the reach as the stream water discharge increased? How did this change in relation to the DOM? Similar to my comment above, I don't think new analysis is warranted – but simply a figure depicting an example of the potential variability of the DOM concentration (DOC and DON) and composition along this 3+ km study reach.*

**AUTHORS:** Following your suggestion, we have included a new figure showing median and percentile concentrations for chloride, DOC, and DON along the reach for both the LLF and non-LLF period (Figure R1, new Figure 2). Statistical significant longitudinal trends in concentration have been indicated with solid lines, and the relative increase in concentration along the reach is indicated in the main text of the results sections (only for significant longitudinal trends).

We believe that this figure and associated results are helpful to describe trends for chloride and DOM along the reach and, further, provides information on how variable stream water concentrations were along the reach. Note that this figure also includes median stream water concentration from tributaries.

The following sentences related to the new Figure 2 have been included in the Results section 4.1 (P7 L23-P8 L5, underlined text):

"During the study period, median Cl$^-$ concentration in the main stream was higher for the LLF (8.6 [7.8, 13.1] [25th, 75th percentiles] mg L$^{-1}$) than for the non-LLF period (7.8 [7.3, 8.8] mg L$^{-1}$) (Mann Whitney test: Z =2.82, df = 1, p = 0.005). Stream Cl$^-$ concentrations increased by 43% and 48 % during the LLF and the non-LLF period, respectively (Figure 2a). A similar pattern was exhibited by riparian GW (Figure S3). In the tributaries, median Cl$^-$ concentration was 10.2 [8.8, 14.2] mg L$^{-1}$. For DOC, median concentration in the main stream was higher for the LLF (843 [643, 1243] μg C L$^{-1}$) than for the non-LLF period (406 [304, 580] μg C L$^{-1}$) (Mann Whitney test, Z =2.55, df = 1, p = 0.008) (Fig. 3a). Stream DOC concentrations increased along the reach by 58% during the LLF period (Figure 2b). In the tributaries, median DOC concentration was 577 [390, 881] μg C L$^{-1}$. For DON, median concentration in the main stream was 58 [35, 78] μg N L$^{-1}$ and showed no seasonal pattern (Mann Whitney test, Z = -0.85, df = 1, p > 0.05) (Fig. 3b). Stream DON concentrations showed no clear longitudinal changes for any of the two study periods (Fig. 2c), though concentrations could vary by 40% on a single date. In the tributaries, median DON concentration was 54 [34, 75] μg N L$^{-1}$. The median stream DOC:DON ratio in the main stream was higher during the LLF (DOC:DON = 22 [14, 43]) than during the non-LLF period (DOC:DON = 8 [5, 15]) (Mann Whitney test, Z = 1.98, df = 1, p = 0.033) (Fig. 3c)."

Moreover, we have built a similar figure for riparian groundwater so that the reader can evaluate by how much chloride and DOM concentration in riparian groundwater varied along the reach (Figure R2). We refer to this Figure in the Results section, but we have decided to include it as Supplementary Material in order to keep the paper as concise as possible and avoid losing focus, which is the stream compartment and in-stream processes.

Regarding the variability of DOM composition along the reach, we would like to highlight that we already provided information in this regard in the former version of the manuscript. In particular, Figure 5 (Figure 6 in the revised version) showed the temporal pattern of the standardized regression coefficient obtained by fitting a linear regression to the values of FI, BIX, and HIX along the reach. Moreover, an additional figure was included in the Supplementary materials showing the same type of calculation for the PARAFAC components (Figure S2, former version of the manuscript).

[Figure]

**Figure R1 (will be new Figure 2).** Longitudinal patterns of (a) chloride, (b) dissolved organic carbon (DOC), and (c) dissolved organic nitrogen (DON) concentrations in stream water along the 3.7km reach. Symbols are median values and whiskers are the interquartile range (25th, 75th percentiles) for the main stream (circles) and tributaries (diamonds). Concentrations are shown separately for the LLF (grey) and non-LLF period (white). Black circles in (b) correspond to the field campaign of the 22 of November of 2010 when DOC concentrations were higher than for the remaining study period. Model regressions are indicated with solid lines only when significant (tributaries not included in the model).

[Figure]

**Figure R2 (will be new Figure S3).** Longitudinal patterns of (a) chloride, (b) dissolved organic carbon (DOC), and (c) dissolved organic nitrogen (DON) concentrations in riparian groundwater along the reach. Symbols are median values and whiskers are the interquartile range (25th, 75th percentiles). Concentrations are shown separately for the LLF (grey) and non-LLF period (white).

Regarding spatiotemporal changes in stream discharge and chloride concentration, note that most of the information requested was included in a previous paper focused on stream nutrient dynamics along the same study reach (Bernal et al. 2015). To avoid redundancies, we first decided to not repeat these results in this paper. Yet, we agree with the reviewer that some additional information about hydrology could be helpful for contextualizing our results. In Bernal et al. (2015), we found that despite stream discharge increased along the reach, area-specific discharge decreased longitudinally. These results indicate that hydrological retention increased at the valley bottom compared to upstream segments. The following sentences have been included in the Study Site section (text underlined) (P3 L30-P4 L2):

"On average, stream discharge increased along the reach from 20 to 70 L s⁻¹. During the study period, the stream gained water in net terms along the reach, yet it lost water towards the riparian zone in some segments, specifically during summer months. Moreover, mean area-specific stream discharge decreased longitudinally, an indication that hydrological retention was higher at the valley bottom compared to upstream segments (Bernal et al. 2015)".

Additional information on hydrology has been incorporated in the text by including Cl⁻ concentration in stream water and riparian groundwater for the two periods (see Table 1 in revised version). Moreover, longitudinal trends in Cl⁻ concentration along the reach for the two periods are now shown in new Figure 2 (this is Figure R1).

**REFEREE:** *Referring to Figure 6, given that the error bars for UDOC and UDON overlap 0, I count 2/10 days where U>0 for DOC and 4, maybe 5 dates where U>0 for DON. I understand the median value is above 0, but given the variability (i.e. the error bars), U = 0 cannot be discounted. I suggest the authors re-cast the results to explain this result and C2 therefore their interpretation more clearly (in reference to text P9 L11).*

**AUTHORS:** We are glad that you have highlighted this issue. Thanks to your comment, we have realized that error bars associated with $U_{DOC}$ values were so large and asymmetric compared to $U_{DON}$ and $U_{Cl}$ because of a mistake when calculating min and max fluxes from the tributaries. After correcting this mistake, former Figure 6 looks as follows:

[Figure]

**Figure 6 (new Figure 7).** Temporal pattern of in-stream net uptake ($U$, either in µg or mg m⁻² h⁻¹) for (a) chloride, (b) dissolved organic carbon (DOC), and (c) dissolved organic nitrogen (DON) at the whole reach scale. Whiskers are the uncertainty associated with the estimation of stream discharge from NaCl slug additions as in Bernal et al. (2015). Values of $U > 0$ indicate that gross uptake prevails over release, while $U < 0$ indicates the opposite. For cases with $U > 0$, the contribution of in-stream net uptake to decrease stream solute fluxes (i.e. $U \times A/F_{in}$, in %) is shown (black bars). The leaf litter fall period (LLF) is indicated.

Following your suggestion, we have recalculated the number of cases for which the stream was acting as a net sink of DOM considering only those cases for which $U > 0$ for both median and error bars. To increase the clarity of our response, we include values of $U_{Cl}$, $U_{DOC}$, and $U_{DON}$ in Table R2. The results showed that the stream acts as a net sink of DOM ($U > 0$) in six (rather than in seven) and in seven (rather than in eight) out of 10 sampling dates for DOC and DON, respectively. In these cases, in-stream processes contributed to reduce stream fluxes by 47 [43, 65] % (rather than by 44 [36, 54] %) and 37 [28, 40] % for DOC and DON, respectively. We have corrected these results in section 4.4 (P10 L2-4).

**Table R2.** In-stream net uptake rates ($U$, either in μg or mg $m^{-2}$ $h^{-1}$) for chloride, dissolved organic carbon (DOC), and dissolved organic nitrogen (DON) at the whole reach scale. The upper and lower limit of $U$ are based on the empirical uncertainty associated with discharge measurements as in Bernal et al. (2015). Values of $U > 0$ (in blue) indicate that gross uptake prevails over release, while $U < 0$ (in red) indicates the opposite.

| | $U_{Cl}$ (mg $m^{-2}$ $h^{-1}$) | | | $U_{DOC}$ (μg C $m^{-2}$ $h^{-1}$) | | | $U_{DON}$ (μg N $m^{-2}$ $h^{-1}$) | | |
|---|---|---|---|---|---|---|---|---|---|
| 27/10/2010 | -3,7 | 0,3 | 4,2 | -132,4 | 162,1 | 480,0 | 9,5 | 34,2 | 58,4 |
| 22/11/2010 | -1,2 | 0,3 | 1,9 | -587,0 | -61,0 | 469,9 | 66,2 | 90,6 | 114,3 |
| 19/01/2011 | -1,3 | 0,1 | 1,5 | -493,3 | -344,7 | -181,8 | 21,5 | 47,7 | 83,5 |
| 01/03/2011 | -1,0 | 0,0 | 1,1 | 265,4 | 348,0 | 433,8 | 9,1 | 19,5 | 30,4 |
| 12/04/2011 | 2,4 | 6,9 | 11,4 | 939,8 | 1287,5 | 1635,2 | 147,5 | 188,5 | 229,4 |
| 26/05/2011 | -1,5 | 0,8 | 3,2 | 24,9 | 189,7 | 403,1 | -43,8 | -17,0 | 20,7 |
| 09/08/2011 | -2,0 | 1,2 | 4,5 | 49,0 | 216,1 | 393,5 | 3,1 | 19,8 | 38,1 |
| 13/09/2011 | 0,1 | 0,8 | 1,6 | -2,9 | 23,7 | 51,4 | -4,5 | -0,3 | 3,7 |
| 26/10/2011 | -1,7 | -0,2 | 1,4 | 39,3 | 205,7 | 383,3 | 18,1 | 24,8 | 32,1 |
| 15/12/2011 | -4,8 | -0,2 | 4,4 | 364,5 | 976,5 | 1596,5 | -75,7 | -42,8 | -9,9 |

**REFEREE:** *Also, did the authors at all consider residence time of the reach in terms of DOM processing (e.g. Casas-Ruiz and others 2017 L&O)? What was the average velocity of the stream reach among sampling dates? And residence time would be a more important factor at base flow than storm flow. While obvious to some, perhaps mention this within the discussion.*

**AUTHORS:** As stated in the former M&M and Discussion, sampling campaigns were conducted during base flow conditions. Thus, the variability of hydrological conditions among sampling dates was not large enough to explore the influence of hydrology on in-stream DOM processing. For example, mean water velocity range from 0.23 to 0.4 m $s^{-1}$ (mean values now included in the Study Site section), while water residence time within the reach ranged from 2.3 to 4.9 h. These values fall within the low range of values reported by Casas-Ruiz et al. (2017) (water residence time from 0.5 to 100 h), where many stream reaches included, or were under the influence of, small reservoirs. Therefore, our data is representative of base flow conditions in running waters of headwater catchments that do not contain natural or artificial water storage structures. Following the reviewer suggestion, we have modified the following paragraph of the Discussion as follows (text underlined) (P11, L5):

> "Yet, our results are representative of base flow conditions which represent ca. 60 % of the annual DOC and DON flux in the study catchment (unpublished data). Moreover, mean water residence time along the reach was relatively low (4h, unpublished data) because running waters predominated and there were no natural or artificial dams along the study reach. Further studies including storm flow conditions and/or reaches with small reservoirs

would be needed to gain a more complete picture of the role of in-stream processes on DOM dynamics and whether headwater streams shift from reactors to pipes with changing hydrological conditions (e.g., Casas-Ruíz et al., 2017; Raymond et al., 2016)".

**REFEREE:** *Overall, the manuscript presentation is clear and concise. Any comments that I may have had regarding grammar or writing was extremely minor (see below).*

**AUTHORS:** Many thanks for your positive comments and grammar suggestions.

**Specific Comments**

**AUTHORS:** All the small specific comments have been incorporated in the main text of the revised version of the manuscript. Mineau et al. (2016) has been included as a reference in the Discussion. Thanks for the language suggestions, they are helpful and we have learned from your explanations. For those specific comments that were more substantial, we include a brief answer below:

P1 L23 – *non-LLF rather than no-LLF*. **OK**
P1 L24 – *I suggest changing 'reflex' to 'reflection' (or reflects) here and throughout the manuscript. Reflex could indicate an opposite or opposing outcome whereas 'reflect' and 'reflection' indicates similarity. Similarity is what I think the authors intend within this context*. **OK**
P2 L18 – *This is very minor, but I would re-cast 'important' as your readers may not understand what is deemed as important in this context. Perhaps recast to 'a significant fraction'?*. **OK**
P3 L6 – *Is there a citation for the carbon vehicle hypothesis?* **OK**
P6 L7 – *The authors calculated uncertainty in U (uptake or generation) based on the variability on stream water Q. Did the authors also consider the variability in the DOM and DON concentrations as well, as flux estimates will vary based both on Q and concentration variability?*
**AUTHORS:** Yes, the reviewer is right: the uncertainty of *U* was based on the error associated with the measurement of stream discharge. Unfortunately, we only collected one stream water sample (from the thalweg) from each sampling site on each field campaign. Thus, we could not consider the empirical error associated with stream water chemistry. Following your suggestion, we have added few sentences in the first paragraph of the Discussion highlighting the uncertainties associated to the mass balance approach (see our detailed responses to reviewer #3 in this regard).

P6 L11 – *for consistency and clarity, instead of referring to 'release' (U < 0) as 'opposite' of uptake – call it 'release'*. **OK**
P6 L17 – *typo – 'steam' to 'stream'*. **OK**
P9 L8 – *typo - 'no' to 'not' statistically significant*. **OK**
P9 L23 – *I suggest re-cast for this latter part of the sentence — minor changes – but perhaps . . . 'stream water and riparian GW to investigate whether stream DOM reflected terrestrial sources or if in-stream processes modified DOM quality.' Elucidate has the same meaning as 'estimate' – but I don't think 'estimate' is the proper word here within the context of this sentence (unless you modify the sentence to 'estimate fluxes' or something similar). Also, see my comment above – 'reflex' should be 'reflect' – and were able to modify – can be simplified to 'modified'*. **OK**
P10, L20 – *The authors already include a number of citations explaining why the uptake rates of DOC and DON were 10-1000 fold lower than rates of in-stream DOM uptake from*

*reported experiments, but I think they should include Mineau and others 2016 – as this particular review paper discusses that ambient DOM uptake «« than DOM uptake using simple sugars, etc. Mineau, M. M., Wollheim, W. M., Buffam, I., Findlay, S. E. G., Hall, R. O., Jr., Hotchkiss, E. R., et al. (2016). Dissolved organic carbon uptake in streams: A review and assessment of reach-scale measurements. Journal of Geophysical Research: Biogeosciences, 121(8), 2019–2029. http://doi.org/10.1002/2015JG003204.* **OK**

P11 L2 – *typo 'tan' to 'than'.* **OK**

P11 L8 – *change 'what' to 'which' reinforces their….* **OK**

**Anonymous Referee #3**

**REFEREE:** *This study investigated the differences and fate of riparian groundwater and in-stream DOC and DON. The study found in-stream production and transformations of DOC that support the assertion that stream corridors serve a key role in biogeochemical cycling of carbon, beyond use as a conduit. The study was well designed and well written. My primary concern was with the lack of context surrounding prior research into hyporheic biogeochemical cycling and the framing of groundwater within the text. The methods describe sampling "riparian groundwater" from a shallow depth near the stream edge. I am not familiar with the term "riparian groundwater," but this sounds like sampling the hyporheic zone of the stream and is quite different from sampling pure groundwater. In addition to context being added to the introduction, I think this distinction needs to be fleshed out in the discussion.*

**AUTHORS:** Many thanks for your positive comments. We are glad that you find "*the study well designed and well written*". Regarding your main concern, we agree that we could be more clear when defining water body compartments. This is, indeed, an essential topic in stream ecology given the difficulty of clearly defining boundaries between the stream, the hyporheic zone and the riparian groundwater.

A priori, riparian groundwater could be considered the water laterally transported from the saturated riparian zone to the stream, while the hyporheic zone would be defined as the subsurface streambed zone that receives water from both the riparian and stream compartments (Bencala et al. 1993). From a hydrological point of view, these definitions become fuzzy because stream water can eventually infiltrate towards the riparian zone. Thus, a more relaxed definition of the hyporheic zone should include the near-stream zone too, because stream water and saturated groundwater can mix along both the vertical and horizontal dimension (not strictly within the streambed zone) (Bencala et al., 2011). In any case, what is clear is that riparian groundwater must traverse both near stream and streambed zones before reaching free flowing waters. From a biogeochemical point of view, groundwater DOM is likely processed along this riparian-stream interface (e.g., Fasching et al. 2015), which makes extremely difficult to identify the exact chemistry of the riparian groundwater entering into the stream (Brookshire et al., 2009).

In this study, and for practical reasons, we assumed that the so call "riparian groundwater" (groundwater collected 1.5 to 2m from the stream channel) was representative of the chemical signature of terrestrial groundwater entering into the stream-hyporheic zone after draining both hillslope and riparian ecosystems. Yet, we cannot rule out that the stream-groundwater mixing front could eventually move towards the riparian zone. Further, we acknowledge that we cannot distinguish whether DOM biogeochemical processing occurred in the stream water column and/or within the hyporheic zone. Thus, and regarding the interpretation of the mass balance results, we agree that it is important to highlight that no distinction could be made between the hyporheic zone and the stream water column.

Following your suggestion, we explicitly refer to the hyporheic zone in the Introduction (P2, L18). Moreover, we have clarified in M&M the definitions and assumptions regarding riparian groundwater as indicated in the earlier paragraph:

> "At each sampling site, we installed a 1 m long PVC piezometer (3cm Ø) in the riparian zone (~1.5 m from the stream channel edge). We assumed this water to be representative of the groundwater entering the stream" (P4 L9),

And:

"Riparian GW must transverse the hyporheic zone before arriving to the stream water column, and thus, we considered that in-stream net uptake was the result of biogeochemical process occurring in both the stream water column and the hyporheic zone" (P5 L30)

Finally, we have included several sentences in the Discussion, acknowledging the uncertainties associated with the adopted mass balance approach (P10, L13-20):

"However, the characterization of the exact DOM chemistry entering from the riparian GW to the stream is a complex issue (e.g. Brookshire et al., 2009). First, the two water bodies (stream and riparian GW) are hydrologically connected throughout the hyporheic zone (Bencala et al., 2011). Thus, hydrological mixing cannot be completely rule out because stream water can eventually penetrate towards the riparian zone (Bernal et al., 2015). Second, DOM in riparian GW is likely processed while traversing the near-stream and hyporheic zones (Fasching et al., 2015). Hence, by sampling only riparian GW (2 m from the stream channel) and free flowing water at the thalweg, we could not distinguish whether in-stream processes occurred in the stream water column, the streambed, or the hyporheic zone".

**Additional comments:**
**REFEREE:** *Consider consistently using "allochthonous" and "autochthonous" to reduce some of the wordiness of describing terrestrial vs. in-stream DOM.*

**AUTHORS:** Thanks for your suggestion. We have carefully read the manuscript and substituted "terrestrial" by "allochthonous" and "in-stream" by "autochthonous" whenever appropriate. Please, see changes made in this regard, in the version with track changes.

**REFEREE:** *The conclusions could benefit from describing directions for future research.*

**AUTHORS:** Following your suggestion, we have included some sentences about future directions in the conclusion section (now named "*Conclusion and future research*") (P13 L30): .

"Further work is needed for disentangling the different mechanism underlying DOC and DON processing within the streams as well as for understanding how environmental factors such as nutrient availability and water residence time drive in-stream DOM processing and changes in DOM quality during different hydrological conditions".

P 12 L21: *Change "modify" to "modifies"* **OK**

**REFEREE:** *Figure 1: I would suggest finding a way to more clearly differentiate between "evergreen oak" and "other." They look quite similar in the key.*

**AUTHORS:** Following your suggestion, have modified the color key of this figure to improve clarity.

[Figure]

**Figure 1 (revised)**. Map of the Font del Regàs catchment within the Montseny Natural Park (NE, Spain). The vegetation cover and the main stream sampling stations along the 3.7-km reach are indicated. Four permanent tributaries discharged to the main stream from the upstream- to the downstream-most site (white circles). The remaining tributaries were dry during the study period. The location of the hillslope springs and soil lixiviates plots is also shown.

**REFEREE:** *Figure 2, 3, 5, 6: You use the same x-axis notation of month/year for all of these plots, but you only list "Time (month/year)" on some of them. I was initially confused by the notation. I suggest adding "Time (month/year)" to the plots that lack it.*

**AUTHORS:** Many thanks. X axis read now the same in former Figures 2, 3, 5, and 6.

**REFERENCES INCLUDED IN THE RESPONSE LETTER**
Bencala, K. E.: A perspective on stream-catchment connections, J. N. Am. Benthol. Soc., 12, 44-47, 1993. doi:10.2307/1467684.

[revised manuscript text omitted]